# Circular Economy Development in the Wood Construction Sector in Finland

**Roope Husgafvel** [1,*] and **Daishi Sakaguchi** [2]

1 Department of Bioproducts and Biosystems, Aalto University, 00076 Espoo, Finland
2 Department of Human Care Engineering, Faculty of Health Science, Nihon Fukushi University, Handa 475-0012, Japan; daishi@n-fukushi.ac.jp
* Correspondence: roope.husgafvel@aalto.fi

**Abstract:** Circular economy development is about a system level change towards enhanced sustainability and circularity covering both biological and technical cycles. This study aimed at exploring, identifying, analyzing and synthesizing the current state of and future outlook on CE development in the wood construction sector in Finland as perceived by various sectoral companies. This study focused on multiple themes related to the importance of the various aspects of a CE and associated approaches in this particular sector. This study applied a qualitative research approach, and a questionnaire survey was the specific method. The survey was sent to both architectural and construction sector companies. This study addressed a gap in research and contributes to the better understanding of the current state of and future outlook on CE development in the wood construction sector. The results indicate that the CE concept is mostly considered to be an important part of building design and construction. However, some respondents found that this concept is new to them. Interestingly, the cascading use of wood and the assessment and measurement of a CE were not at all familiar to many respondents. Particularly important CE aspects in the wood construction sector include, for example, sustainability and the long life cycles of products, components and materials; co-creation and cooperation covering the whole life cycle of construction and the whole supply chain; training and competence development; and design for a CE, sustainability and long life cycles. Many essential elements of a CE were in use, coming into use or in consideration by many of the sectoral companies.

**Keywords:** circular economy; wood construction; Finland

## 1. Introduction

CE development aims at keeping products, components and materials at their highest utility and value at all times, covering both biological and technical cycles [1]. In addition, it has been noted that there is a clear relationship between the CE and the UN SDGs [2] and that CE goals are directly or indirectly linked to the achievement of the UN SDGs [3]. It is also noteworthy that CE transition needs to be aligned with both sustainability (environmental, social and economic) and sustainable development goals, including necessary sustainability assessments to guide businesses and industry, considering the fact that circularity does not necessarily lead to sustainability [4]. In general, the transition to a CE implies the closing and slowing down of resource loops to maintain the highest possible economic value of products, components and materials and minimize the environmental impact [5]. Additionally, choices that contribute to a sustainable future and enhance a fair society focused on well-being are also major elements of CE development [3].

A transition to a CE is important because of its multiple positive impacts at the global level such as reductions in resource and energy use, greenhouse gas emissions and waste generation [6–14]. In general, the current economic linear growth model is critically challenged by the finite nature and growing scarcity of nonrenewable resources; the increasing stress on renewable natural resources; and the crossing of planetary boundaries [15]. Therefore, CEs are about a means to an end that is driven by the global need to transition to a

circular model due to the crossing of planetary boundaries and the destabilization of the Earth's systems [16]. A CE includes focuses on, e.g., rethinking value creation and the use of all materials [1], and it is often driven by alternative business models; regulatory trends; technological development and advances; resource constraints; the degradation of and pressures on natural systems; and socio-economic opportunity [1,15].

There are many benefits and opportunities associated with the CE concept and approach, such as (1) a focus encompassing economic and industrial renewal on the realignment of economic success metrics with social realities and a shift of perspective from value chain improvement to meeting human needs [17]; (2) a transition towards a low carbon economy [18]; (3) addressing global challenges [19] such as climate change, the loss of biodiversity and the overuse of natural resources [20]; (4) building natural, economic and social capital [21]; and (5) the production of economic well-being within planetary boundaries and a reduction in the use of natural resources [20].

Design is a major part of a CE and, particularly, in the construction sector. Therefore, it is important to keep in mind aspects related to design for circularity and sustainability, including ecodesign concepts and approaches. Eco-design is about the integration of environmental considerations into product design and development [22]. In addition, it can be an important driver for industry and companies provided that potential barriers related to a lack of eco-knowledge are addressed [22]. Product design strategies are very important for both the implementation of and overall transition towards a CE [23]. In addition, eco-design is particularly important in the building sector due to significant environmental pressures related to resource consumption and waste generation [24].

The design strategies to promote eco-design include, for example, Design for Circular Economy; Material Efficient Design; Design for Disassembly; Design for Waste Minimization; Design for Maintainability; and Design for Adaptability [24]. It has been noted that more focus is needed on eco-designs that encompass, for example, design strategies that emphasize materials substitution; structural optimization; and reductions in resources and energy consumption [23]. The tools for the eco-design of buildings include, for example, life cycle assessment and green building certification schemes [24]. In addition, it is noteworthy that there can be eco-misperceptions among eco-designers about the environmental sustainability of design solutions regarding life cycle of products and the selection of the appropriate level of detail in eco-design and eco-assessment [25]. Additionally, there are multiple barriers that limit the implementation of eco-design in the building sector such as a lack of appropriate legislation; a lack of knowledge among designers; and a lack of applicable tools and methods [24].

The relationship between designs and business models is important because the performance of a business model can be combined with a design for a technological cycle and the product's final life cycle, which means that materials can be easily reclaimed and recycled within a circular economy if the product has a good design [26]. For long-life products and recovered resources, businesses cannot operate without an appropriate strategy and design. Recent changes in business models could result in the end of the linear economy and the start of circularity of the product. While this might seem a daunting prospect at first, it also increases the importance of the design for a circular economy [27].

Nationally, sustainable products and services; longer material cycles; innovations; digital solutions; and the sustainable use of renewable natural resources all play a major role in CE development [3,20]. Additionally, the construction sector plays a major role in CE development in Finland and focuses on, e.g., low carbon CE solutions and significant reductions in environmental impacts [3]. The national CE goals in Finland highlight the importance of ecosystems (e.g., innovation), education and skills for CE development, including a particular emphasis on life cycle thinking and a focus on the whole life cycle of buildings in the context of construction [3].

CE development is strongly promoted in the EU. The European Green Deal, for example, aims at transforming the EU economy for a sustainable future encompassing the integration of sustainability into all policies with a particular emphasis on a CE that

includes resource- and energy-efficient building and renovation [28]. The measures of the EU CE Action Plan to promote a CE in the construction and building sector encompass the promotion of circularity principles throughout the lifecycle of buildings; the sustainability performance of construction products; a reduction in greenhouse gas emissions; material recovery and efficiency; durability and adaptability; and life cycle assessment [29]. CE issues in the wood construction sector that are linked to EU level policy goals encompass, e.g., the cascading and sustainable use of materials such as wood [30–38].

There are also new developments, such as the New European Bauhaus projects, that add a creative and cultural dimension to the European Green Deal and aim at promoting sustainable innovation, technology and economy [39]. The New European Bauhaus projects highlight the importance of long-term and life-long thinking within the industrial ecosystem with a focus on (1) a CE and circularity; (2) circular and sustainable design and architecture; (3) the use of sustainably produced and procured nature-based building materials (e.g., wood); (4) the life extension, reuse, regeneration and transformation of existing buildings; (5) recovered and renewable materials (e.g., use and design); and (6) designs for sustainability and new business models [39].

The EU CE Action Plan also encompasses sustainability principles such as (1) product durability, reusability, upgradability and repairability including measures to address hazardous chemical in products; (2) increasing recycled content in products; (3) remanufacturing and high-quality recycling; (4) incentives for products with a high level of sustainability performance; (5) reductions in environmental and carbon footprints; (6) the digitalization of product information and product-as-a-service models covering whole life cycles; and (7) restrictions related to single-use, premature obsolescence and the destruction of unsold durable goods [29]. In addition, the EU Forest Strategy highlights the sustainable use of wood in accordance with the cascading principle and the CE approach with an emphasis on long-lived circular products and materials that provide the highest value for carbon storage and CE [40].

The overall aim of this study was to explore, identify, analyze and synthesize the current state of and future outlook on CE development in the wood construction sector in Finland as perceived by various sectoral companies.

## 2. Materials and Methods

This study applied a qualitative research approach [41], and the chosen specific method was a questionnaire survey [42–50], which was applied as an online survey that was sent via email directly to respondents. This study is important because it addresses a gap in the research and contributes to an enhanced understanding of CE development in the wood construction sector as perceived by both design-oriented and construction-oriented experts. The overall approach was to explore CE development in this sector via themes and issues raised in previous studies; the literature; and various documents in this particular field of study such as strategic and policy initiatives. In addition, our own previous research related to both a CE and the construction sector in particular was duly considered in the formulation of the themes and questions. In brief, the survey themes, questions and answering options were based on (1) the literature review and previous studies; (2) our previous studies; and (3) the assessment of the global, EU and national operational environment for CE development in the construction sector with a special emphasis on wood construction. In addition, face validity (peer review) [48] was applied to check the quality of the whole questionnaire.

The survey themes encompassed (1) the importance of the CE concept as a part of building design and construction; (2) the familiarity of CE aspects in the construction sector; (3) the importance of the main principles of a CE in the built environment; (4) the importance of CE aspects in the design of wooden buildings; (5) the importance of CE aspects related to wood materials, components and products; (6) the importance of CE aspects in the wood construction sector; (7) the importance of approaches to integrate CE into wood construction; (8) the use or introduction of approaches to promote CE in the

design of wooden buildings; (9) the importance of aspects related to the cascading use of wood for wood construction; (10) the importance of aspects related to CE ecosystems in the wood construction sector; (11) the importance of CE business models and associated aspects in the wood construction sector; (12) the importance of approaches to assess CE performance in the wood construction sector; and (13) the use or introduction of approaches to promote the CE of wood construction.

The applied survey software was Webropol, and the questions were written in the form of "how important is/are"; "how familiar are with"; and "are the following in use or will you introduce them". The survey was anonymous and voluntary, and its respondents included architects and company representatives (e.g., managers, project managers, designers and experts) in the construction sector. The questionnaire was sent to respondents in architectural companies (150 in total) and in construction companies (150 in total) via email. All the respondents received the same survey and associated questions. The response rate was 8.7% (n. 26) covering 8 construction companies; 7 architectural companies; 6 manufacturing companies; 2 contractor companies; 1 company focused on projects; and 2 companies in other fields (Figure 1). The sizes of the responding companies are presented in Figure 2.

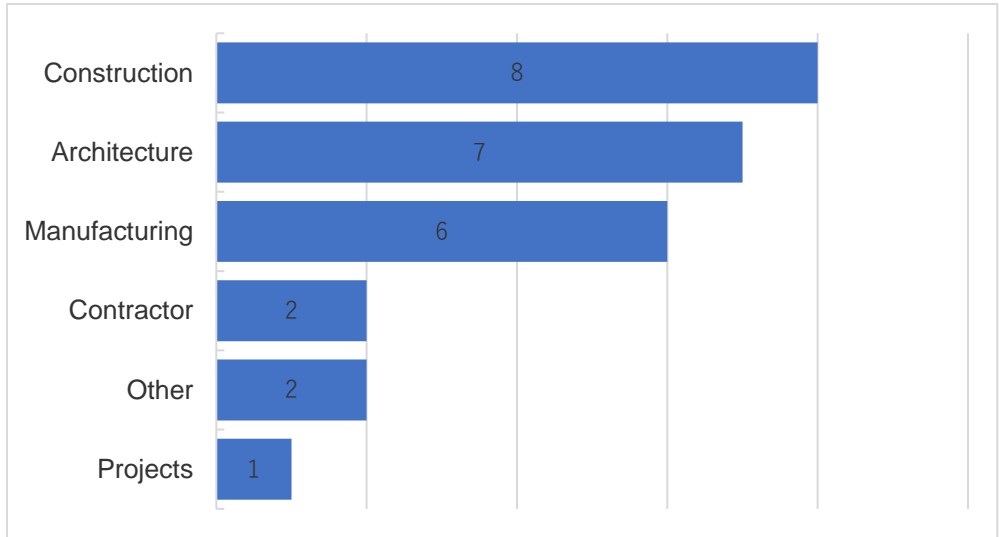

**Figure 1.** Field of business.

The questionnaire was structured and formal with a selection of multiple closed questions. Each question encompassed multiple elements and the respondents were asked to make a selection between (1) "very important", "important", "partly important" or "not at all important"; (2) "in use", "will be introduced", "in consideration" and "not in use or will not be introduced"; or (3) "very well", "well", "somewhat" and "not at all". One question also included "the concept is new to me" as an option. The results are presented as figures based on the main themes and associated survey questions. The chosen style is to include the number or percentage of respondents in the columns.

In the survey, cascading use of wood was defined as the full utilization of the full potential of wood resources via sequential and multiple uses encompassing, e.g., recovery, recycling, remanufacturing, refurbishment and/or reuse in multiple new applications aiming at the maintenance of highest possible utility, usability and value at all times/uses (covering longest possible use as a material instead of combustion). Ecosystem was defined as a whole that consisted of a network of multiple actors, which is created around a common goal and by which broad system level results can be achieved.

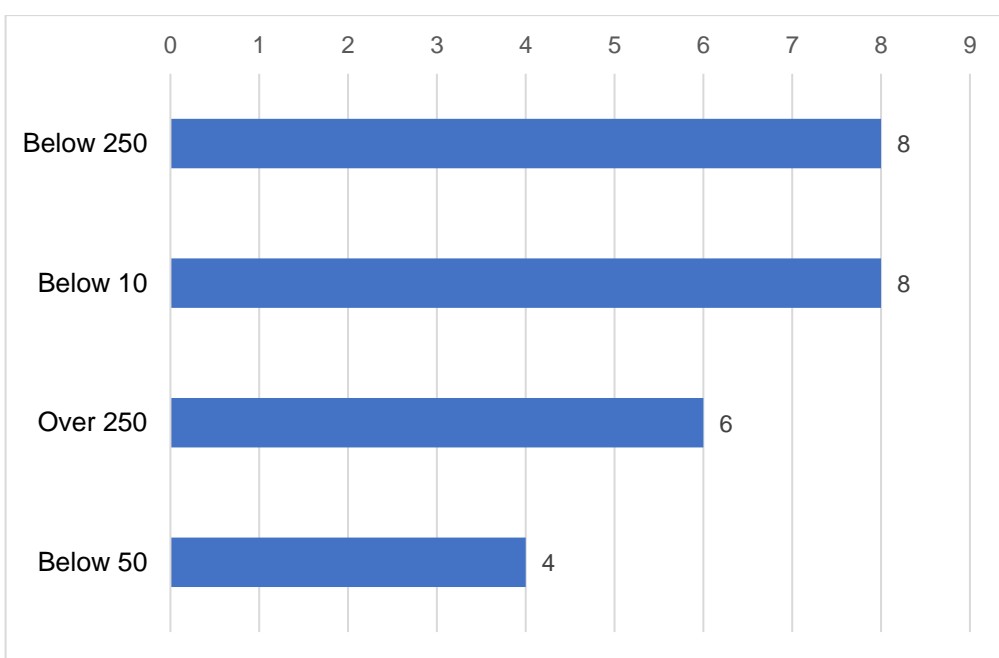

**Figure 2.** Size of company.

## 3. Results

### 3.1. The Importance of the CE Concept as a Part of Building Design and Construction

The results indicate that the CE concept is mostly considered to be very important or important or at least partly important part of building design and construction (Figure 3). However, some respondents found that this concept is new to them. In Finland, CE has been an important societal focus area including, for example, strategy and policy development, international events and specific sectoral initiatives (e.g. construction sector). Thus, the perceived importance of the concept may have been promoted by these continuous efforts.

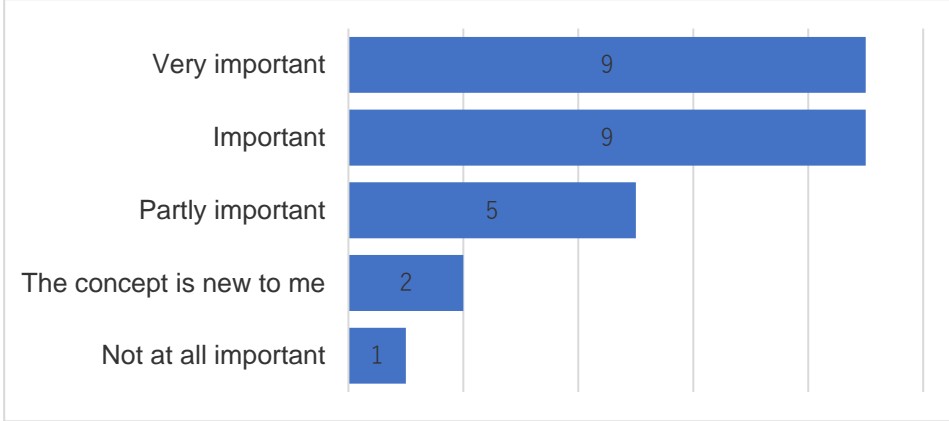

**Figure 3.** The importance of the CE concept as a part of building design and construction (number of companies).

It has been noted that the lack of knowledge about the environmental performance and associated benefits of the various building design and construction strategies is a major gap that prevents CE uptake within the industry [51]. The architectural design firms need to address sustainability during the design process including the enactment of measures to address challenges associated with the construction supply chain such as raising the awareness of all parties about sustainability in the design phase [52]. In addition, they need to consider training and resources to enhance the design and technical skills of architects

and design managers as well as the integration, coordination, communication and sharing of information within the supply chain [52]. Furthermore, research is needed on new design typology to assess the environmental performance of each of the different design and construction strategies in relation to the CE concept (whether the strategies minimize building-related environmental impacts) [51].

### 3.2. The Familiarity of CE Aspects in the Construction Sector

The results indicate that the respondents are most familiar with systems and life cycle thinking and renewable raw materials, whereas the cascading use of wood, products as services and sharing platforms, and the assessment and measurement of CE were not at all familiar to many respondents (Figure 4). The respondents were quite familiar with the design aspects of CE. Similarly to the results in Section 3.1, the continuous efforts to create awareness of and to advance a CE in Finland, including sectoral initiatives, have enhanced the familiarity of companies with the concept and associated key aspects.

Previous studies have identified numerous important points about both CE development and sustainability in the construction sectors, such as (1) the contribution of the adoption of a CE in the construction industry to achieve the UN SDGs covering environmental, social and economic aspects [53]; (2) the need for the construction industry to be more sustainable due to its major impact on the environment and the importance of improving the performance of the construction supply chain to promote the achievement of sustainability objectives [52]; (3) the need for both legal and financial motivators to implement a CE in the construction sector [54]; and (4) the fact that a CE can significantly improve the sustainability of the building sector [55]. A CE should be integrated into all phases of construction, and all professionals should be trained to recognize and utilize circularity potential in the construction industry [53].

Currently, a CE is not appropriately implemented in the building sector, and there are major barriers to its implementation, including the lack of environmental laws and regulations; the lack of support from public institutions; and the lack of public awareness [13]. It has been noted that the building industry is in the early stages of developing CE practices [11] and that most buildings are not designed in accordance with the CE principles [56]. In addition, there is no general agreement on what circularity means for the construction sector or on the required steps to achieve circularity [57]. Popular circular strategies in this sector encompass recycling, prefabrication and selective demolition, whereas design for disassembly, closed-loop recycling and design in layers have low adoption [58]. A CE in the building industry can be advanced via the application of a CE framework to the prefabricated building sector with a focus on reusability; adaptability (e.g., the second use of components); reduction; and the recyclability of components [59].

Currently, legislation does not sufficiently promote reuse and resource management, and there is also a lack of attention to the building level in CE policies and to the evaluation of environmental sustainability in circularity strategies [7]. Barriers to the adoption of strategies that are aligned with a CE in the construction industry include upfront costs and budgets; project schedules and timelines; current business models; the lack of awareness; and regulation [58]. The promotion of CE and the elimination of waste in the construction sector could focus on, e.g., modular building design; the creation of new building materials; the recycling of building materials; the reuse of construction and demolition waste; the renting of unused spaces; and the co-use of equipment [15].

In addition, the promotion of a CE in the construction and real estate sector requires a whole value chain approach to the involvement of stakeholders in the design, decision-making and project development phases that considers the integration of life cycle sustainability assessment approaches [6]. The adoption of circular practices in the construction sector requires collaboration among all relevant stakeholders, including general agreement on what circularity means and the support of technology development to promote deconstruction and disassembly [57]. Formal institutions, such as legal rules and regulations, play a key role in motivating both companies and individuals towards a CE in the construction

sector by supporting the coordination of stakeholders and potential beneficiaries to fully realize the benefits of a CE [54].

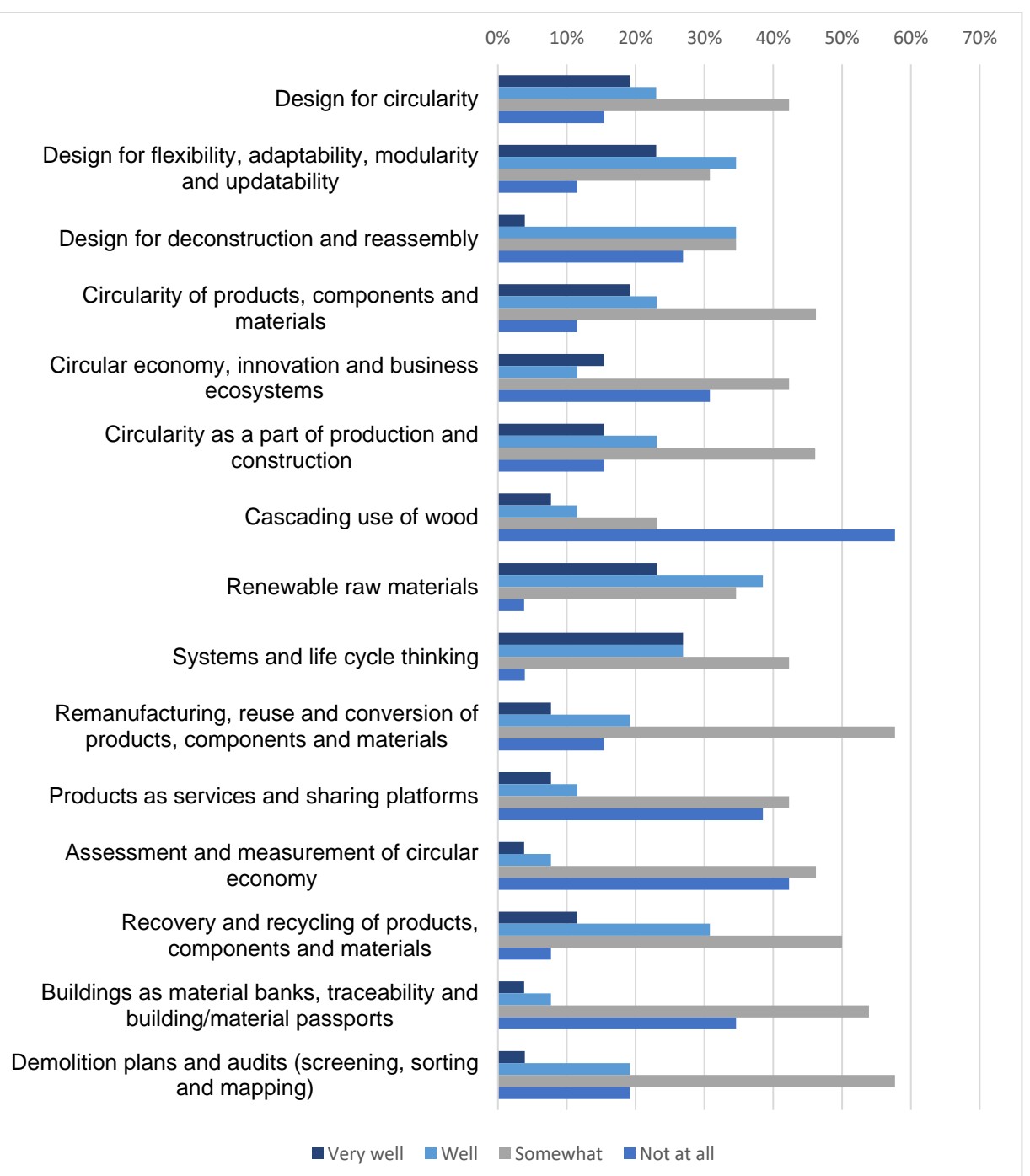

**Figure 4.** The familiarity of CE aspects in the construction sector.

### 3.3. The Importance of the Main CE Principles in the Built Environment

The results indicate that the reduction in energy consumption; prevention and minimization of waste generation; and sustainability and long life cycles of products, components and materials were considered to be very important by the respondents (Figure 5). Many aspects of CE were also considered to be important such as new business models; circularity as a part of production and construction; the use of renewable raw materials; and the remanufacturing, reuse and conversion of products, components and materials.

It may partly explain the importance of energy and waste aspects that they are already familiar with various contexts such as resource and energy efficiency compared to the relative new CE concept.

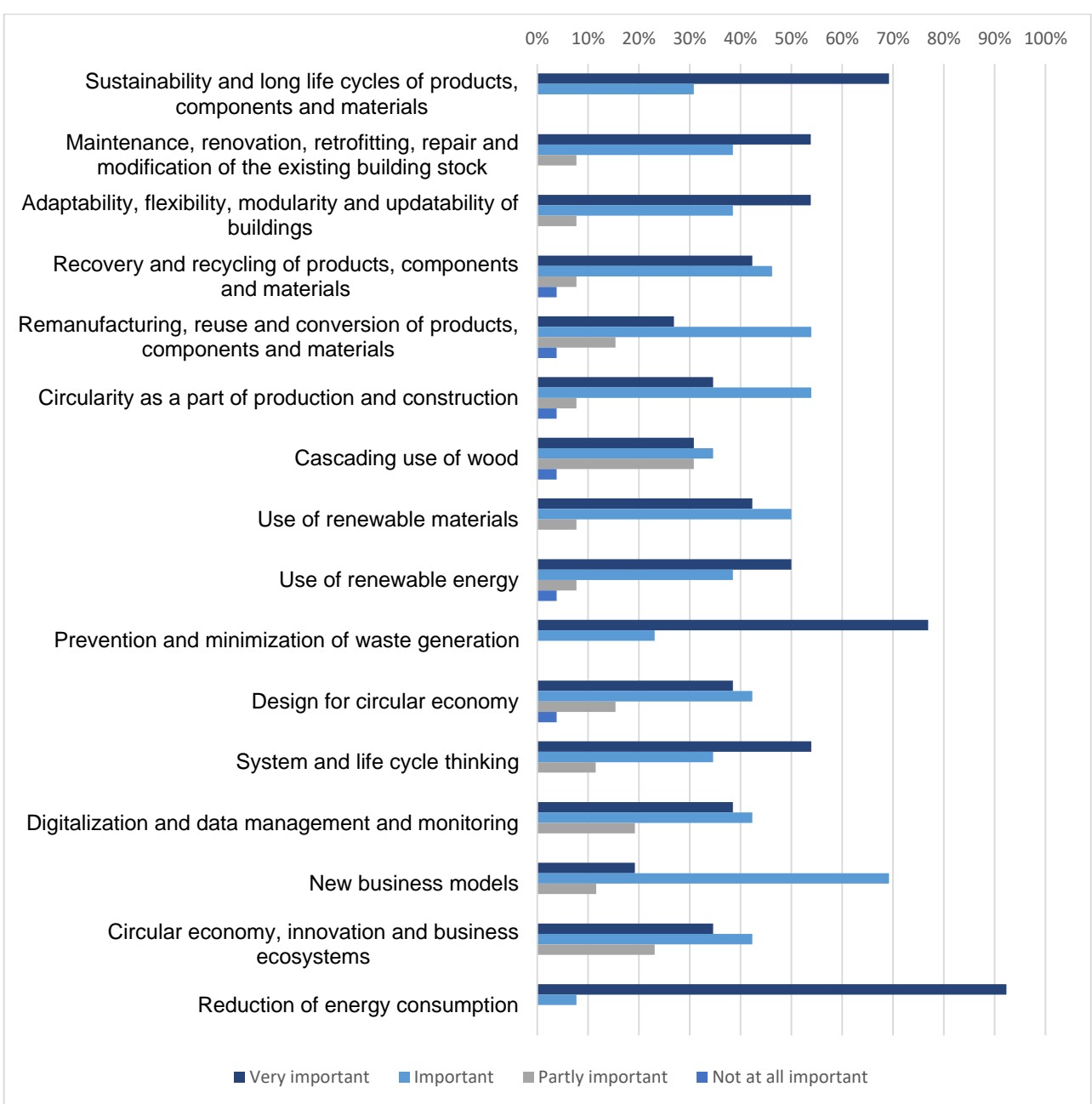

**Figure 5.** The importance of the main CE principles in the built environment.

Previous studies have recognized that (1) academia and industry should work together to promote the adoption of CE principles in the built environment, including the exploration and development of tools to advance a CE and sustainability [60]; (2) the CE concept in the built environment including digital tools and material passports provides opportunities for value creation and innovation [61]; (3) a CE provides many benefits and circular opportunities for the built environment in cities such as the scaling-up of reuse and recycling in construction and demolition waste; design for flexibility and energy efficiency; digitalization; and shared spaces [21]; (4) a CE can help to turn waste into wealth via, e.g., a focus on wasted resources, wasted life cycles and the capabilities of products and wasted embedded values of materials and components [15]; and (5) a CE provides a fundamental

means for achieving sustainability and carbon neutrality provided that it is supported by appropriate product design and a new collaboration between stakeholders in the value chain and business models [62].

The advancement of a CE in the construction sector and making buildings and the built environment more sustainable requires an emphasis on (1) circular business models; (2) integration between stakeholders in the value chain; (3) circular supply chains; (4) best strategies and tools in the early design stages; (5) and government support such as laws, incentives and subsidies [14]. Understanding the circular value of materials and systems in the built environment requires reliable information provided by material passports covering the whole supply and value chain [61]. There are also tools for converting wastes into resources such as raw materials passports, materials banks and markets for salvaged materials [60]. Material passports and digital technologies are useful CE tools to manage material flows and to decarbonize the built environment [61]. In addition, they can help to increase the value of materials; enable access to sustainable materials; promote reverse logistics; and create incentives for manufacturers and suppliers to produce circular materials [61].

### 3.4. The Importance of CE Aspects in the Design of Wooden Buildings

The results indicate that the communication between designer and both client and constructor; designs for sustainability and long life cycles; the use of renewable energy; the minimization of the use of chemicals and hazardous substances; and co-creation and cooperation covering the whole life cycle of construction and the whole supply chain were considered to be very important by the respondents (Figure 6). In addition, design aspects of a CE; the cascading use of wood; performance, quality and content criteria for recycled and reused materials, components and products; the use of sustainable and certified materials, components and products; and building information modeling and the visualization of all environmental impacts were considered to be important by many respondents.

Previous studies have acknowledged that circular product design requires focus on (1) products, components and materials that retain their economic value for as long as possible and minimizes their environmental impact [5]; (2) a significant reduction in the use of natural resources and environmental impacts [56]; (3) the implementation of the principles of CE by extending the lifespan of products considering reuse, repair and refurbishment aspects and design for assembly/disassembly, maintainability, remanufacturing, recycling, sharing economy and sustainable behavior [63]; (4) the application of a life cycle perspective considering resources used in production, processes at the end of use cycle, social and behavioral aspects, complex value chains with multiple stakeholders and environmental impact associated with manufacturing, use and recovery processes [5]; and (5) the assessment of the environmental performance of each of the different design and construction strategies in relation to the CE concept [9,51].

The idea of circular product design is to loop used products, components and materials back into the economic system [5]. In addition, designs for a CE can encompass designs for durability, product-life extension, modularity, disassembly and reassembly, recycling, standardization and compatibility, and accessibility [5]. There is also a need to look at how LCA can be applied as an integral component of the design process and not only as an additional aspect [64] and how important CE focus areas at the building level such as design/designs for reversible buildings are addressed [7].

In general, a CE can be promoted and environmental impacts can be reduced via (1) designs for disassembly, durability, adaptability and low-impact materials; (2) strategies that go across and beyond the life cycles of buildings, components and materials (e.g., future reuse); (3) material recovery and recycling; (4) the correct selection of materials; (5) the substitution of short-lived and embodied greenhouse-gas-emission-intensive materials; (6) and the reuse of existing buildings, components and materials [10].

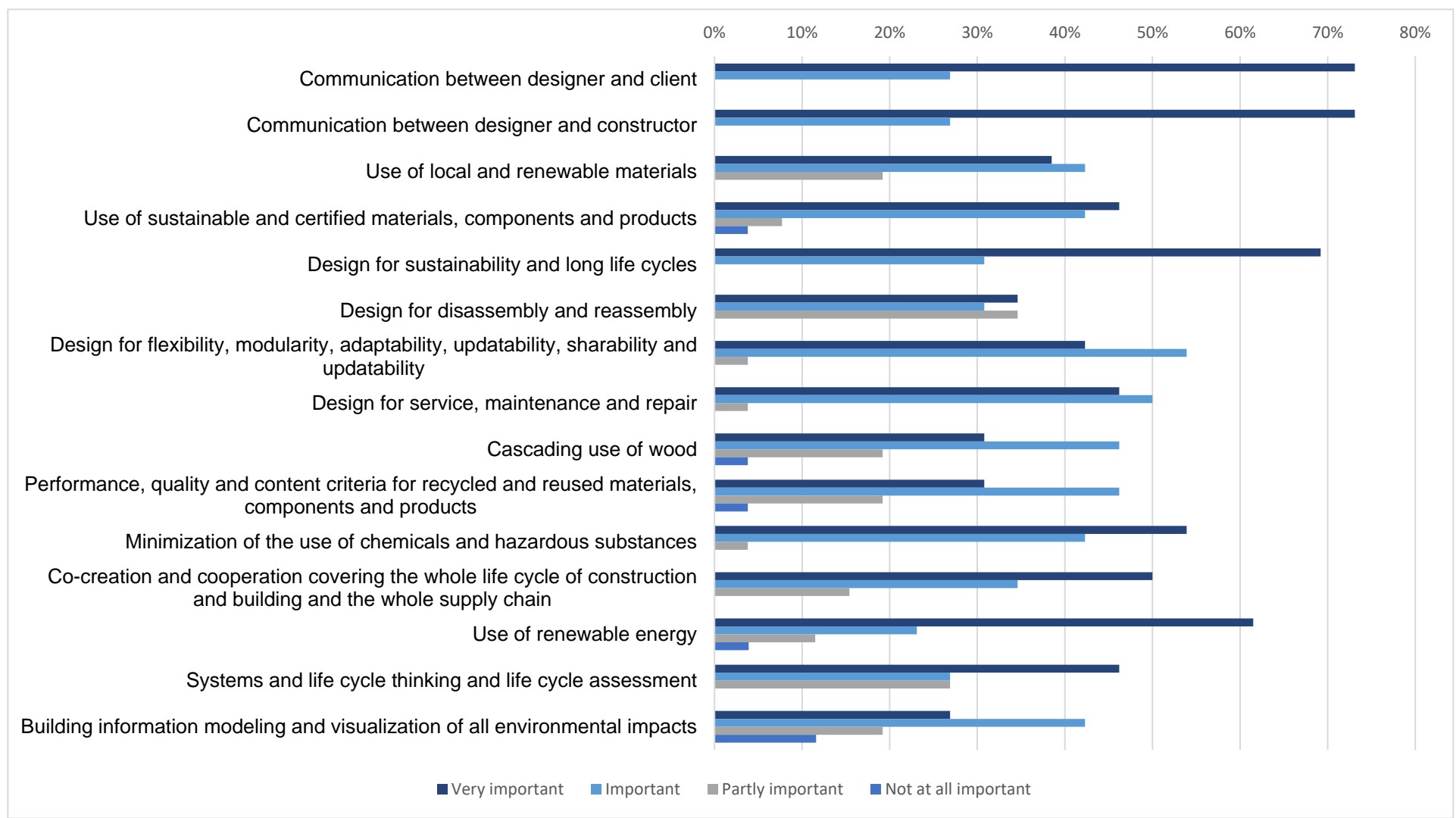

**Figure 6.** The importance of CE aspects in the design of wooden buildings.

### 3.5. The Importance of CE Aspects Related to Wood Materials, Components and Products

The results indicate that the prevention and minimization of waste generation; sustainable forest management, forest certification and certified wood products; renewable raw materials; and recovery and recycling were considered to be very important by the respondents (Figure 7). Many CE aspects were considered to be important such as design for circularity; rethinking and reduction in material use; the sustainability of raw materials; and systems and life cycle thinking.

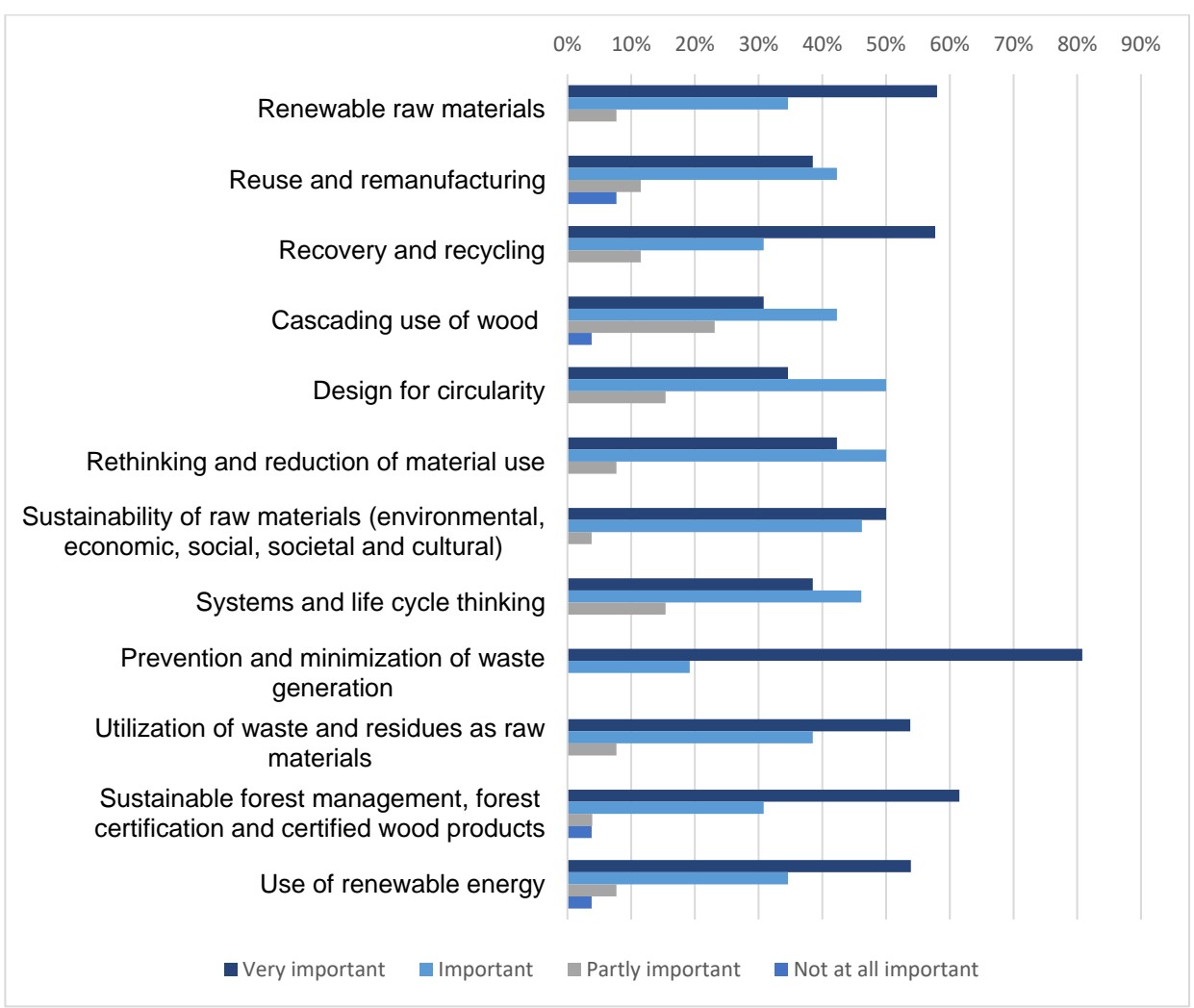

**Figure 7.** The importance of CE aspects related to wood materials, components and products.

Our previous findings indicate that the best approaches to promote CE encompass the use of sustainable and renewable raw materials; consumer awareness; and the design, use, and manufacturing of sustainable, recyclable, reusable and repairable products, components and materials [65]. In addition, the development of CE and sustainability-oriented products, components and materials is essential [65]. Sustainable, long-lasting and fixable products and new services and products are among the important drivers and opportunities associated with CE development [66]. Important focus areas also encompass overall wood construction aspects; sustainability; life cycle thinking, management and assessment; the recycling and reuse of products; material and energy efficiency; reduction in greenhouse gas emissions [67–69].

Interestingly, product-based approaches, climate change and local industrial symbiosis received very little attention [68]. Collaboration among various actors; the more efficient use

of raw materials; the utilization of by-products and side flows; and international guidelines and best practices were also among the issues that were considered important [69]. In addition, it has been noted that the adoption of a CE at the buildings and materials levels can be promoted by (1) the circular design of buildings (e.g., reusable and disassembly focused design and flexible and adaptable design); (2) a reduction in the demand for buildings and materials (e.g., reuse of materials and components, refitting of existing structures and refurbishment); and (3) circular business models (circular supply chain, recovery strategy and maintenance, collaborative consumption and service and product-as-service models) [60].

Wood can be a sustainable resource when harvested and managed responsibly, but it is important to consider the specific practices and sources of wood products that can ensure their sustainability. Wood construction can be an environmentally sustainable alternative to traditional construction methods [37]. Using wood as a building material can also have a lower carbon footprint than using materials such as concrete or steel. Wood also has natural insulating properties, which can reduce energy consumption in buildings. Additionally, wood construction can help promote sustainable forest management practices, which can help ensure that forests are maintained for future generations.

The sustainability of wood as a construction material requires the consideration of environmental impacts related to both forest management and end-of-life scenarios [70]. In addition, the use of bio-based materials such as wood in the construction sector significantly contributes to the mitigation of climate change [70]. The reduction in the carbon footprint of a building requires that a large amount of wood is used in its construction [71]. Carbon storage in the building sector can be enhanced by the large scale combination of sustainable bio-based buildings materials and prefabrication [70].

A CE of timber is not often addressed or implemented, and there is a need for enhanced material recovery to prolong the life cycle of wood, including the associated storage of biogenic carbon [72]. Timber-based construction leads to a carbon stock outside forests [73], and the sustainable use of wood may help to address material resource challenges and to replace energy-intensive building materials [74]. Carbon storage targets can be best achieved by the maximum application of wood materials to replace fossil materials, and wood should be produced in a sustainable way and used in the best possible way (e.g., cascading use) [73]. In brief, sustainability and a long lifespan of products, components and materials are important for the future development of CE [66].

### 3.6. The Importance of CE Aspects in the Wood Construction Sector

The results indicate that training and competence development; sustainability and long life cycles of products, components and materials; the maintenance, renovation, refurbishment, retrofitting and conversion of existing buildings; the sustainability of materials; market creation for recovered and recycled materials, components and products; and co-creation and cooperation covering the whole life cycle of construction and the whole supply chain were considered to be very important by the respondents (Figure 8). In addition, a CE, innovation and business ecosystems; designs for CE and sustainability; and the planning and auditing of demolition were considered to be important by many of the respondents.

Our previous findings indicate the following: (1) the development of CE requires focus on both sustainability and the long lifespan of products, components, and materials [66]; (2) construction and buildings are among the priority focus areas that offer the best opportunities to promote sustainable and CE-oriented public procurement in the future, which encompasses the need to update old buildings and spaces to combat climate change [75]; (3) public procurement can promote a CE by acknowledging the whole product chain and life cycle as well as the obligatory recycling of products, components and materials and a focus on the whole product life cycle [66]; (4) public procurement can advance a CE via sustainability and CE criteria [75]. In general, reusability, renewability and long-lived wood products are topics that are directly linked to the goals and principles of CE such as

extending product lifespans, importance of regenerative systems and reusability, which is one of the 10R principles of CE.

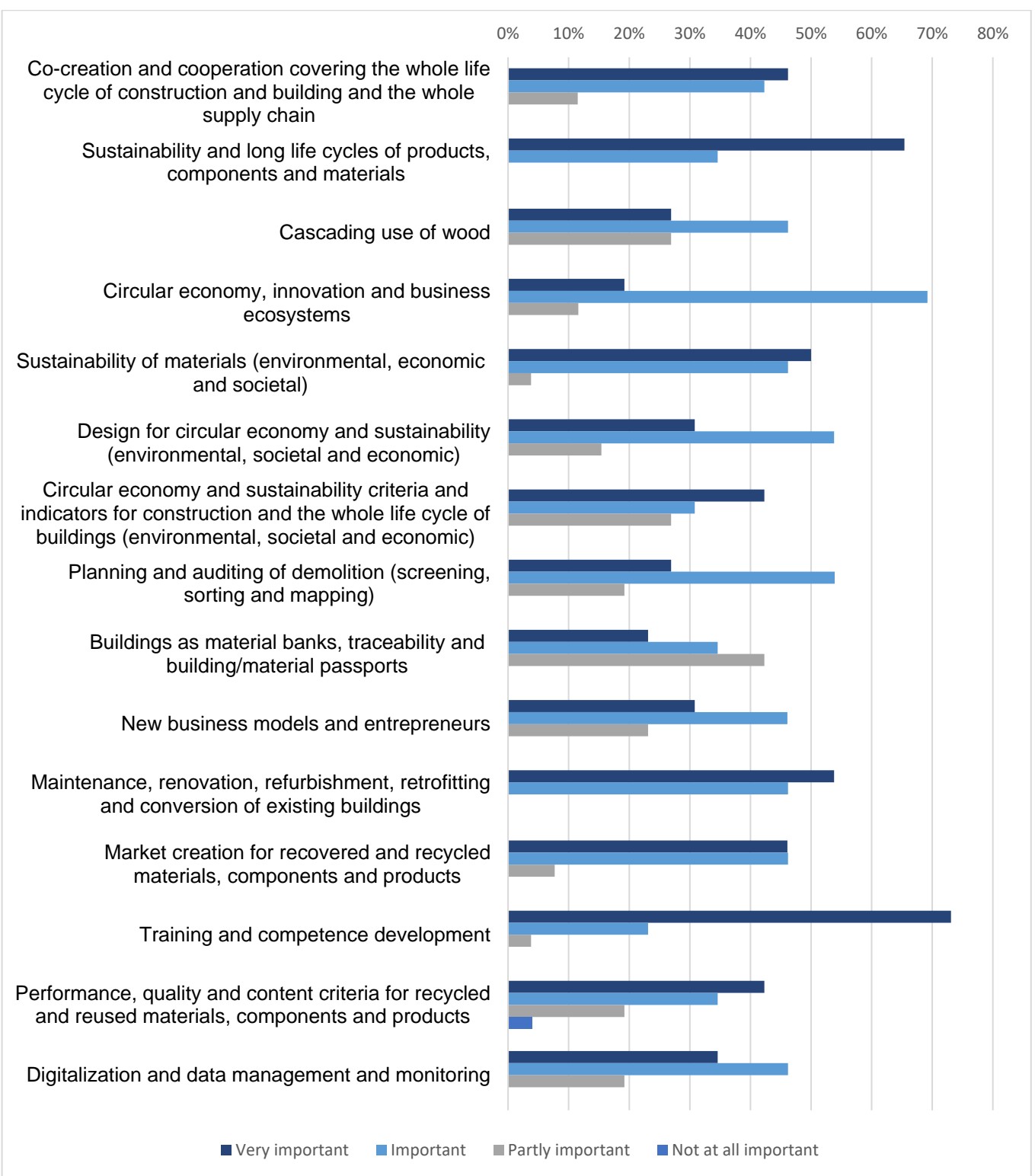

**Figure 8.** The importance of CE aspects in the wood construction sector.

CE has multiple implications for both practice and policy related to various forest products and associated end uses [57]. However, a CE in timber is still not put into practice or considered in the construction sector, even though it is technically possible to achieve

the reuse of structural timber [72]. Important focus areas related to the use of wood in construction include, e.g., design; building service and material life; local materials; and transport and energy aspects [76]. Multiple measures are needed to realize the potential for a more sustainable building stock including, e.g., policy instruments and incentives for the use of innovative wood construction materials [77]. The sustainability aspects of wood as a construction material, such as renewability; the reduction in environmental impacts and resource use; positive well-being and human health aspects; reusability; carbon storage in long-lived products; climate change mitigation; and an increase in the overall carbon stock, have been recognized in multiple previous studies [35,72,76,78–82].

Previous studies have also recognized the following: (1) wood can be a highly sustainable building material because it is naturally renewable, reusable and recyclable, and it also has good strength-to-weight ratios and acoustic and thermal insulating properties that make it useful for multiple applications in buildings such as structural beams/frames, windows, wall/flooring materials, door frames and furniture [83]; (2) overall sustainability in the construction sector can be enhanced by the use of wood and timber structures, including the creation of environmental, social and economic benefits [84]; (3) wood is an important construction material that plays a major role in the context of both a CE and a bioeconomy [30]; (4) the use of renewable materials (e.g., wood) in buildings could contribute to more sustainable construction and make it a part of a bioeconomy [85]; and (5) the sustainability of wood as a building material depends on various issues such as appropriate forest management (sustainable forest management and forest certification standards); manufacturing methods; site assembly; the use of glues and other chemicals; and transport distances [83].

### 3.7. The Importance of Approaches to Integrate CE into Wood Construction

The results indicate that the reduction in the use of chemicals and dangerous substances; waste prevention and the minimization and utilization of waste and residues as raw materials; the extension of product life cycles; and training and competence development were considered to be very important by the respondents (Figure 9). Many approaches were considered to be important such as demolition plans and auditing; CE management, assessment and reporting; and buildings as material banks, traceability and building/material passports.

Our previous findings indicate that (1) the best way to create and maintain value in a CE is the design and manufacturing of sustainable, recyclable, reusable and repairable products, components and materials [65]; (2) important CE drivers and opportunities include sustainable, long-lasting and fixable products and new services and products [66]; (3) the best approach to promote sustainable production is the use of renewable raw materials, whereas the best approach to promote sustainable consumption is the enhancement of consumer awareness [65]; and (4) the areas with the most potential to create significant CE innovations encompass sustainable, recyclable, reusable and repairable products, components and materials and sustainable and renewable raw materials [65]. In general, a CE aims at maximizing the value of products in each point of its life, and it changes economic logic as it highlights methods of sufficiency (not production), including reuse, recycling, repair and remanufacturing as well as the preservation of physical stocks [18].

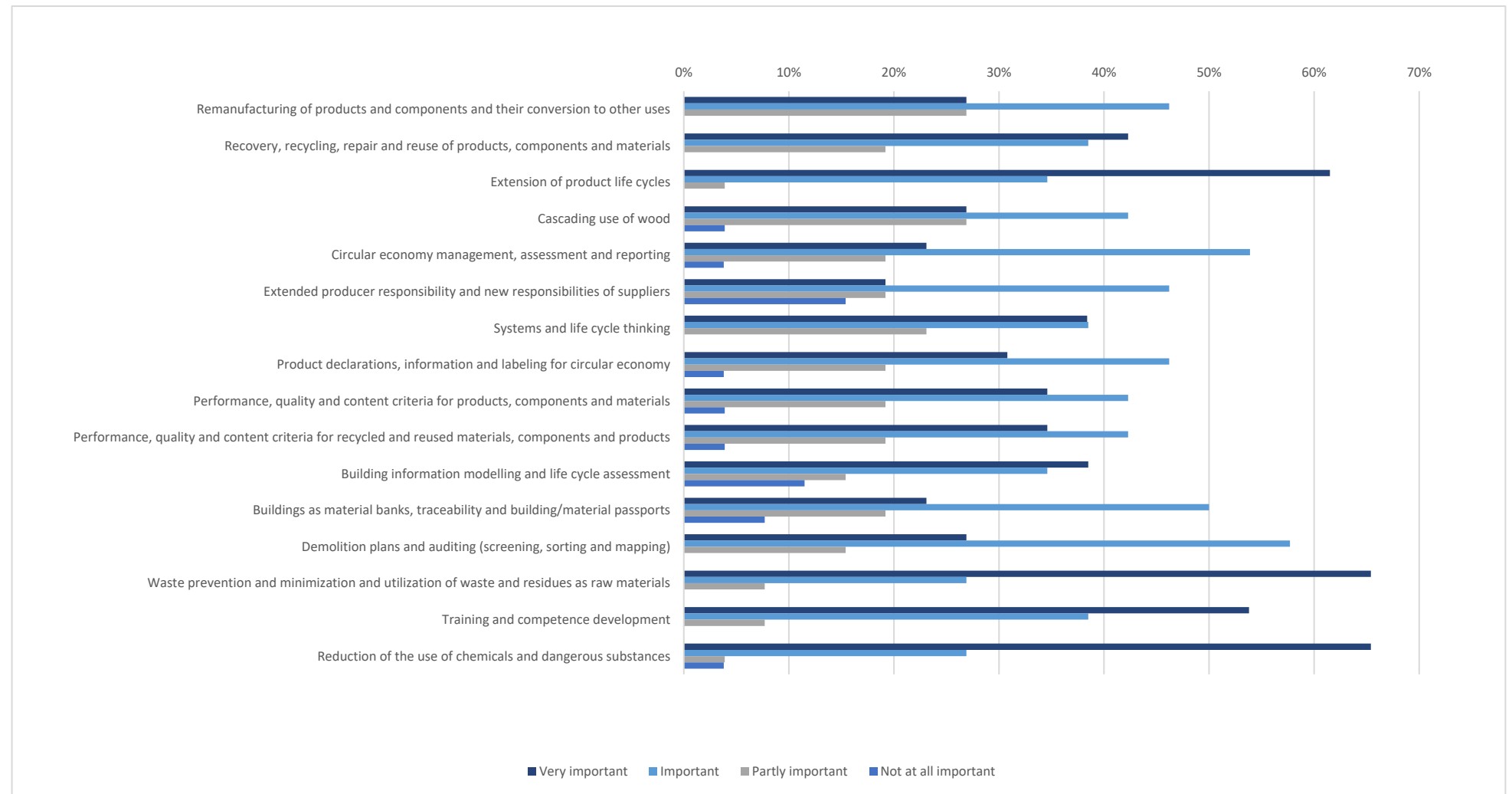

**Figure 9.** The importance of approaches to integrate CE into wood construction.

The development of circular buildings, including circular supply chains and the enhanced reuse of building materials, can be promoted by the establishment of a material market place for long-lived products, components and resources (easy exchange between demolition sites and (re)development projects) and the adoption of take-back schemes by suppliers for short-lived products [86]. The enabling factors that promote the transition to a CE model in the construction industry include education and cultural change; data availability; policies and incentives; and novel voluntary stewardships [58]. Interestingly, low-tech design and approaches have been noted to be very promising for the further development of circular design and the overall advancement of a CE and sustainability in the building sector with a focus on, e.g., reversible building concepts, the circular and reversibility potential of buildings, local materials, resource circularity, material recovery and recycling, and technological innovation [87–89].

Previous studies have also recognized that (1) potential CE practices in the building industry include the design of materials for circularity; the use of reused, up-cycled or recycled materials; and the application of digital tools such as BIM to enhance the transparency of material use in buildings [11]; (2) focus is needed on demolition and waste planning (e.g., sorting, reuse and recycling) considering the potential roles of environmental and construction product declarations, BIM and material/building passports, new economic incentives for companies and new taxation structures [90]; and (3) CE development in the construction and real estate sector requires a comprehensive and circular perspective on all life cycle phases of buildings with a special focus on the service life and reuse/recycling phases (buildings and building materials) considering the application of social life cycle assessment [6].

Recently, more focus has been placed on CE strategies (reuse, repair, refurbish, recycle and recover) and concepts (e.g., related to material loops) [51]. In addition, it has been noted that a transition towards a sustainable circular renovation building process requires a change in policies (e.g., the integration of circular practices into green public procurement and incentives for design for disassembly, reuse, recycling and the use of secondary materials); relationships (e.g., the networking of operators to implement circular strategies and the training of experts); and specific tools to promote sustainability and circularity [91]. Low-impact biomaterials; multiple-use and after-use cycles; longer use based on adaptable design; and resource efficiency are also important in the context of components with a long functional–technical lifespan [9].

### 3.8. The Use or Introduction of Approaches to Promote CE in the Design of Wooden Buildings

The results indicate that sustainability and the long life cycles of products, components and materials; the maintenance, renovation, refurbishment, retrofitting and conversion of existing buildings; and the sustainability of materials were in use by many of the respondents (Figure 10). In addition, training and competence development; designs for a CE and sustainability; a new CE, innovation and business ecosystems; and the sustainability of materials were coming into use according to many of the respondents. Interestingly, many respondents highlight long-term sustainability with particular emphasis on life cycle thinking, encompassing a consideration of all phases from materials to renovation and retrofitting.

Many important approaches such as buildings as material banks, traceability and building/material passports; CE and sustainability criteria and indicators for the whole life cycle of construction and buildings; and digitalization and data management and monitoring were in consideration. Performance, quality and content criteria for recycled and reused materials, components and products; market creation for recovered and recycled materials, components and products, and new business models and entrepreneurs were not in use or coming into use according to many of the respondents.

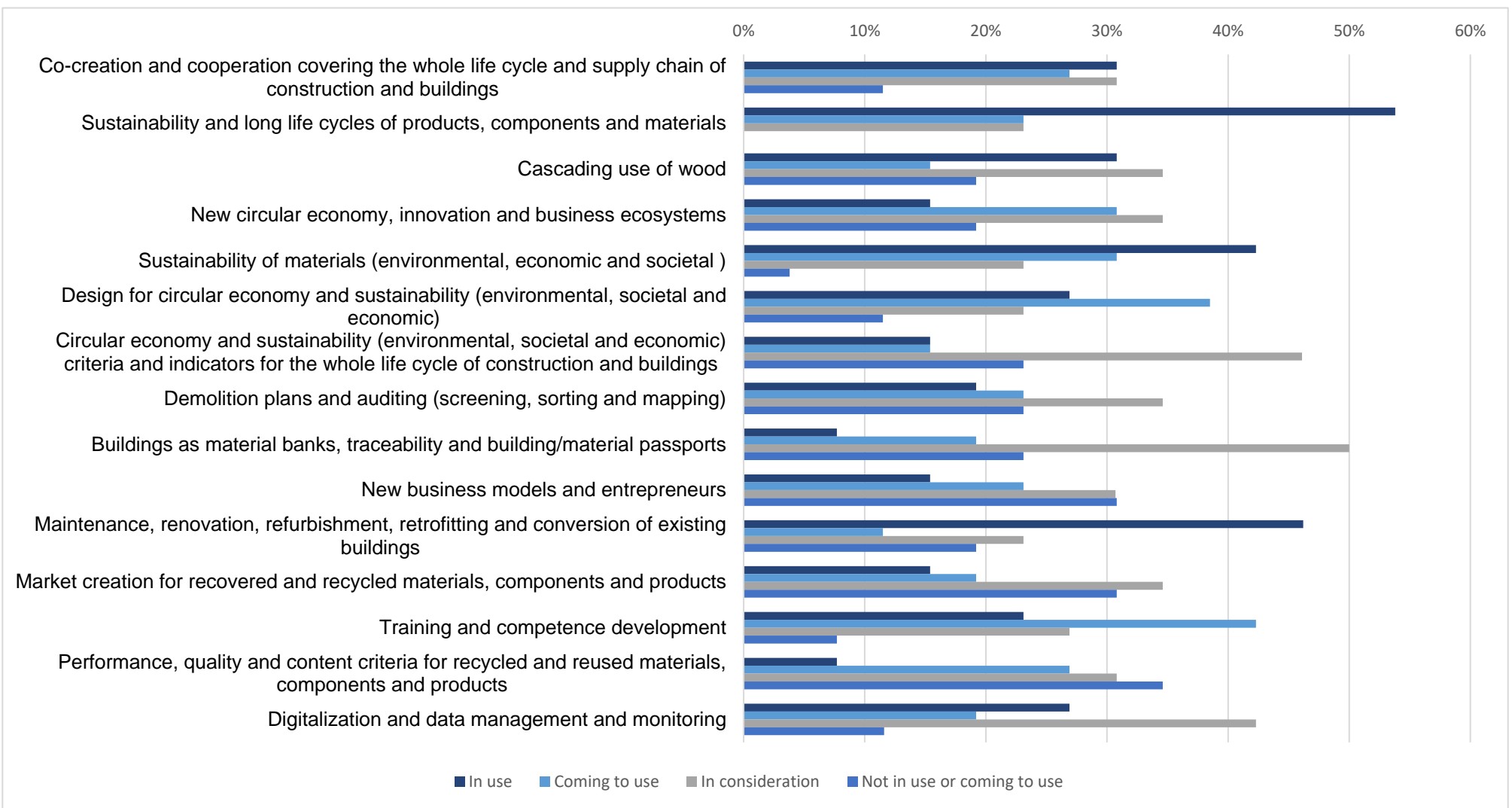

**Figure 10.** The use or introduction of approaches to promote CE in the design of wooden buildings.

Previous studies have recognized the following: (1) architects can promote the use of engineered wood products in building due to, e.g., climate benefits, low environmental impacts, aesthetics and fast construction, and there is a need to increase the knowledge, information, experience and influence over material selection of the architects in this field [92]; (2) architects and structural engineers perceive that their influence on material selection is weak regarding the use of structural timber in multi-story construction and that developers and contractors play a more significant role in material selection [93]; (3) architects do not necessarily have a significant impact on the selection of building materials, and other decision makers involved in building projects may prefer other materials than engineered wood products [92].

For example, carbon stock and climate change mitigation effects can be increased by enhanced cascade use of harvested wood products (e.g., long-lived products) including the development of new designs and material technologies [80]. Barriers to the increased use of wood products in multi-story residential buildings include, e.g., the mismatch in influence and material preferences (frame material) and conflicts of interest (cladding material) [94]. The use of wood among architects and structural engineers is also limited by knowledge gaps and weak support from the wood industry [93]. It has been noted that the large-scale recovery of lumber is not possible because buildings are not designed for disassembly and deconstruction. The use and service life of woody biomass (limited resource) should be optimized [35].

The circular design of products, components and materials can be achieved by designing them for a long product life and enabling effective repair, refurbishment, remanufacturing and recovery, and the recycling of parts [5]. Important focus areas for the promotion of circular building design include, e.g., definitions, methods and tools for architects to support informed decision-making (e.g., the choice of the correct materials and information on the environmental impacts of buildings) [56].

In addition, designs for deconstruction can help reduce waste and preserve the value of materials and resources over time. In building design, designs for deconstruction can involve traditional wood joint systems that allow for the easy disassembly and reconfiguration of building components, such as walls and floors. It can also involve using materials that can be easily separated and recycled [95]. Designing for deconstruction is also one strategy to promote a CE and a global sustainability agenda, but there are barriers to this approach such as a lack of (1) strict legislation and policies; (2) adequate information at the design stage; (3) a sufficient market for recovered components; (4) effective tools and difficulties related to the development of business cases [96].

### 3.9. The Importance of Aspects Related to Cascading Use of Wood for Wood Construction

The results indicate that reduction in the use of chemicals and dangerous substances; enhanced sustainability and longer life cycles of products; recovery and recycling of products, components and materials; full utilization and realization of the potential of all wood resources and their multiple and sequential use; and training and competence development were considered to be very important by the respondents (Figure 11). Many aspects were considered to be important such as a CE, innovation and business ecosystems; market creation for recovered products, components and materials; new business models and entrepreneurs; designs for a CE and cascading use; and the definition of highest utility and value options for the use of wood resources from both societal and private perspectives. Interestingly, many respondents seem to be familiar with the practical challenges related to the use of chemicals and dangerous substances as it was perceived as a very important aspect to be addressed.

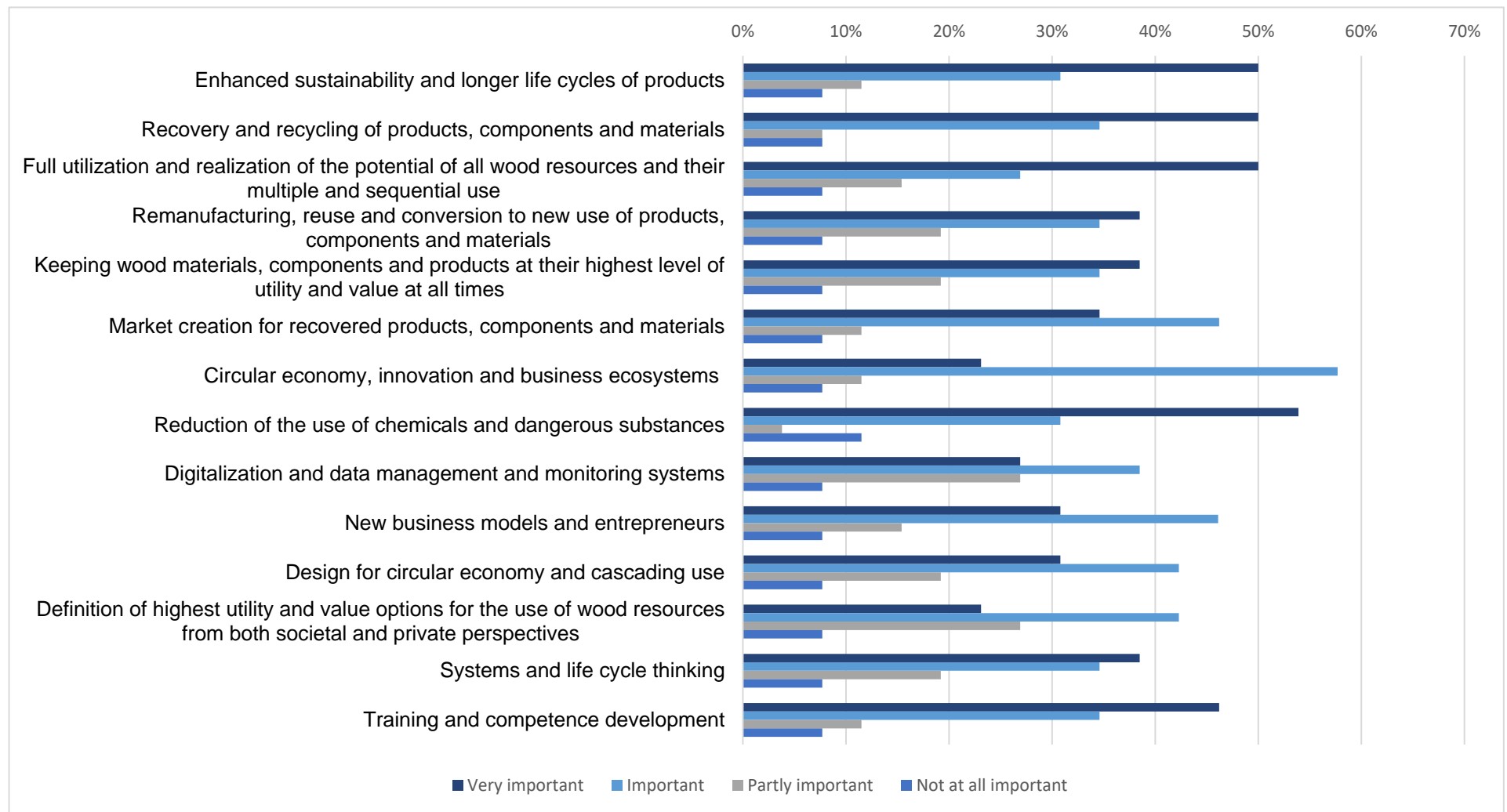

**Figure 11.** The importance of aspects related to cascading use of wood for wood construction.

In general, cascading use refers to the multiple uses of the wood resources from trees including, e.g., the use of recovered and recycled resources and residues [36]. Resource cascading is about the sequential exploitation of the full potential of a resource during its use, and it provides a way to promote the more efficient use of raw materials [97]. Wood from the demolition of buildings can be recovered with relatively good condition and has potential for cascading in different ways such as being used for other building components, furniture or wood products [98]. In general, a CE of wood that includes cascading and the reuse of structural timber as well as prolonging the life cycles of construction materials can contribute to the creation of significant environmental benefits and substitute virgin raw materials [72].

In addition, the cascading use of wood contributes to the mitigation of climate change [72,74,99,100], including, e.g., the reduction in greenhouse gas emissions [73,74,97]. The establishment of a market for second life wood products; the reduction in the demand for virgin wood; and waste generation all require comprehensive knowledge [101]. Our previous research suggests that the cascading of solid wood and materials requires demand from the construction sector/other customers or incentives/legal requirements [67].

There is a significant potential wood resource that is available for cascading in the near future in Finland, and there is a need for further research on the quality, type and future availability of recovered wood products to advance effective reuse and recycling [101]. The selection of the highest possible value option in the context of cascading is influenced by a societal or private point of view that focuses on overall sustainability or a single impact category [31]. In addition, cascading utilization clearly fits into the holistic CE framework, which includes resource management with a strong focus on bio-based materials and utilization possibilities for resources such as reuse, recycling and up-cycling [31].

Previous studies have recognized the following: (1) the sustainability of wood use can be improved by cascading, and the advancement of cascading requires focus on appropriate decision-making and, particularly, on the business case for cascading [102]; (2) cascade use should be evaluated in a comprehensive manner to identify its potential direct and indirect environmental and ecosystem impacts [31]; (3) the cascading use of wood can contribute to the reduction in both global warming and the use of primary wood as well as improve the performance of the whole wood utilization system [103]; (4) there is an increasing potential for the use of recovered timber material in the future encompassing the cascading of a large dimension of solid wood in the construction sector [104]; and (5) wood cascading promotes a more intensive use of limited biomass resources and helps to replace more carbon and energy intensive non-wood materials [105].

Cascaded use is a major driver of value creation in the CE context along with, e.g., extended product life, reuse, repair and maintenance [1]. In addition, the increased use of wood and cascading can contribute to the reduction in environmental impacts [100], such as impacts related to the use of wood resources and the use of wood for energy [99], and to the improvement of resource efficiency by substituting other materials and products [74]. Currently, the cascade principle and practices are not fully established, and it is difficult to determine the benefits of cascading [31]. There is also a need to consider changes in the legal guidelines to promote cascading (e.g., the reuse of structural components that meet strength/property and contamination requirements) [37].

In addition, research has not typically addressed policy limitations related to wood cascading in the context of a CE [106]. The potential amount of waste wood for cascading is significantly higher that currently utilized, and there is a considerable amount of recovered wood from the building sector available for high-quality material cascading, high-value recycling and secondary utilization [37]. It has been noted that (1) the cascading of renewable resource wood can significantly contribute to the increase in the overall lifetime of resource wood, significant savings in primary resource use and the overall use of renewable resources if applied on a large scale [97]; (2) cascading is encouraged via legislation, and the utilization of waste wood in cascades (the use of untreated wood in material applications) is a preferable option to maximize benefits and to minimize environmental impacts [103];

(3) the cascading use of biomass also contributes to the development of bioeconomy [99]; and (4) the cascade utilization of biomass can contribute to the more efficient use of biomass, which also creates more socio-economic advantages and reduces the potential associated negative impacts (e.g., on biodiversity) [107].

Previous studies have identified many interesting and noteworthy points about cascading: (1) increased wood use efficiency (e.g., cascading use) contributes to reduced pressure on virgin wood, the availability of wood biomass for other uses and reduced harvest pressure in forest ecosystems, including avoided emissions associated with industrial wood harvest [99]; (2) the feedstock for the cascade production of new materials (e.g., particleboard) is growing, and material recovery needs to be prioritized over energy recovery via cascading [30]; (3) wood cascading requires a focus on longer lifetimes of wood products to increase climate change mitigation potential (biogenic carbon flows) and on waste wood management and processing and material recycling to reduce environmental impacts [100]; (4) the cascading use of wood encompasses a consideration of, e.g., time, value and function aspects [73]; (5) collection and sorting need to be improved to steer waste wood streams into the best possible applications and to promote the utilization of recovered wood in cascading [108].

In addition, the cascading framework in the context of CE could benefit from the consideration of the integration of cascading with the R-imperatives (e.g., reuse) associated with a CE; the governmental dimension for organizing material allocation; and the governance and steering perspective [109]. The cascading of wood should focus on keeping solid recovered wood in as large dimensions as possible because each degradation in size (e.g., chipping) leads to diminished possibilities for future utilization and because wood materials based on small dimension wood often requires the use of resins and other chemicals [110]. Important focus areas related to wood cascading potential encompass the amount, dimensions and quality of the future recovered timber [104].

It has been noted that (1) the entire wood use cycle and the overall recycling efficiency should be taken into consideration in the context of cascading and associated principles [35]; (2) wood cascading requires a focus on sustainability assessment, carbon balancing and governance of long-lived products; product design; and production (e.g., patterns) and consumption (e.g., changes in consumer behavior) sides of waste mitigation [106]; (3) wood should be cascaded multiple times, including its repeated reuse in the highest possible quality (e.g., wood construction elements are recycled to particle boards) [100]; (4) wood product cascading and the cascading use of woody biomass require a focus on preconditions such as eco-design, the traceability of materials and improved collection and sorting, as well as on forest ecosystem protection [99]; and (5) detached houses show high potential for cascading in terms of quantity [101].

For example, the high-quality wood cascade of wooden beams is a very promising recycling option to reduce environmental impacts [100], and it is technically possible to manufacture, e.g., laminated wood products from recovered wood [102]. Extending the material life of wood over multiple product lives also extends carbon storage time and creates multiple benefits from a unit of wood [110]. Important focus areas related to cascading include, e.g., a better knowledge of wood waste composition and quality and an improvement of both recycling routes and sorting techniques [30]. In addition, the promotion of CE of wood/timber also requires a specific focus on (1) guaranteeing strength and safety in the context of the reuse of structural timber; (2) standardized assessment criteria that are needed to guarantee mechanical properties and to ensure the structural safety of buildings; and (3) the prolonging of the life cycle of wood considering the associated storage of biogenic carbon and the need to prioritize material recovery instead of energy recovery [72].

Cascading in the wood sector can be described based on a combination of (1) product reuse; (2) material recycling; and (3) the substitution of fossil/mineral products, and each of these aspects has impacts on $CO_2$ reduction, carbon storage and resource productivity and efficiency targets [73]. The cascading of high-quality recovered wood in large dimensions

that are free of contamination from the construction sector (e.g., solid beams) should first focus on the production of smaller dimension timber (e.g., lamellas) for another service time, followed by chipping and further cascading as particle- or fiberboards [110]. Wood cascading can be promoted by, e.g., wood banks (maintaining the availability of wood with certain properties at a certain time), the banning of waste wood dumping and the eco-taxation of resources [97].

Post-recovery options for wood lumber include, e.g., its reuse as lumber, its reprocessing as particleboard and pulping. Furthermore, the carbon and energy balances of the cascade chains of recovered wood lumber are influenced by land use effects (alternative possible land uses when less timber is harvested) such as (1) carbon storage in unharvested biomass; (2) substitution effects (reduced demand for non-wood materials); and (3) direct cascade effects (properties and logistics of virgin and recovered materials) [105]. Recovered lumber could be, e.g., processed into wood panels despite the general lack of end-of-life strategies [57]. The realization of the full potential of cascading use requires a mix of approaches based on local conditions to overcome barriers, such as cleaning recovered waste wood (technical); the lack of integrated approaches to material and energy applications of biomass (governance); and a dependence on upstream products (market) [33].

A more holistic assessment of wood cascading is needed, which includes a focus on resource efficiency; realistic cascading systems; the influence of multiple cascading steps on resource efficiency; consequential LCA; and the implementation of a product level resource efficiency indicator that considers various categories/resources [111]. For example, wood flow analysis can be applied to the quantification of cascading at the market level covering sectors and products, and current cascading use is mainly focused on recovering post-consumer wood in accordance with a CE and resource efficiency initiatives [33].

*3.10. The Importance of Aspects Related to CE Ecosystems in the Wood Construction Sector*

The results indicate that cooperation between all actors covering the whole life cycle of buildings and construction; co-creation and design; training and competence development; and buildings as material banks, traceability and building/material passports were considered to be very important by the respondents (Figure 12). In addition, ecosystems for information and know-how; new business models and entrepreneurs; ecosystems that promote new entrepreneurship; the identification and creation of new and surprising connections between actors; and digitalization, platform ecosystems, building information modeling and data management were considered to be important by many of the respondents.

Our previous findings indicate the need to address technological, economical and social barriers to the CE such as (1) the lack of general knowledge about circular economy opportunities and seeing the "big picture"; (2) the few economic benefits associated with sorting and recycling (profitability); (3) sustainability marketing and consumer awareness; (4) the development of sustainable, recyclable, reusable and repairable products, components and materials; and (5) the creation of economic value and incentives [65]. For example, wooden multi-story construction products are sustainability-driven and multi-actor projects that require collaboration and learning that can be advanced via the business ecosystem approach [112].

CE strategies in the construction sector need to overcome limitations related to policies, definitions and projects including (1) the too narrow scale for the application of the CE concept (e.g., multiple scales such as city, region, country, EU or international beyond the site/project); (2) the lack of coordination between the CE and spatial planning (e.g., the need to integrate planning into CE action); (3) the low consideration of the territorial context in the definition of strategies (e.g., local resources and issues); and (4) the uncertain scope of the CE (e.g., balancing broader sustainable development and more operational approaches to the CE concept) [113].

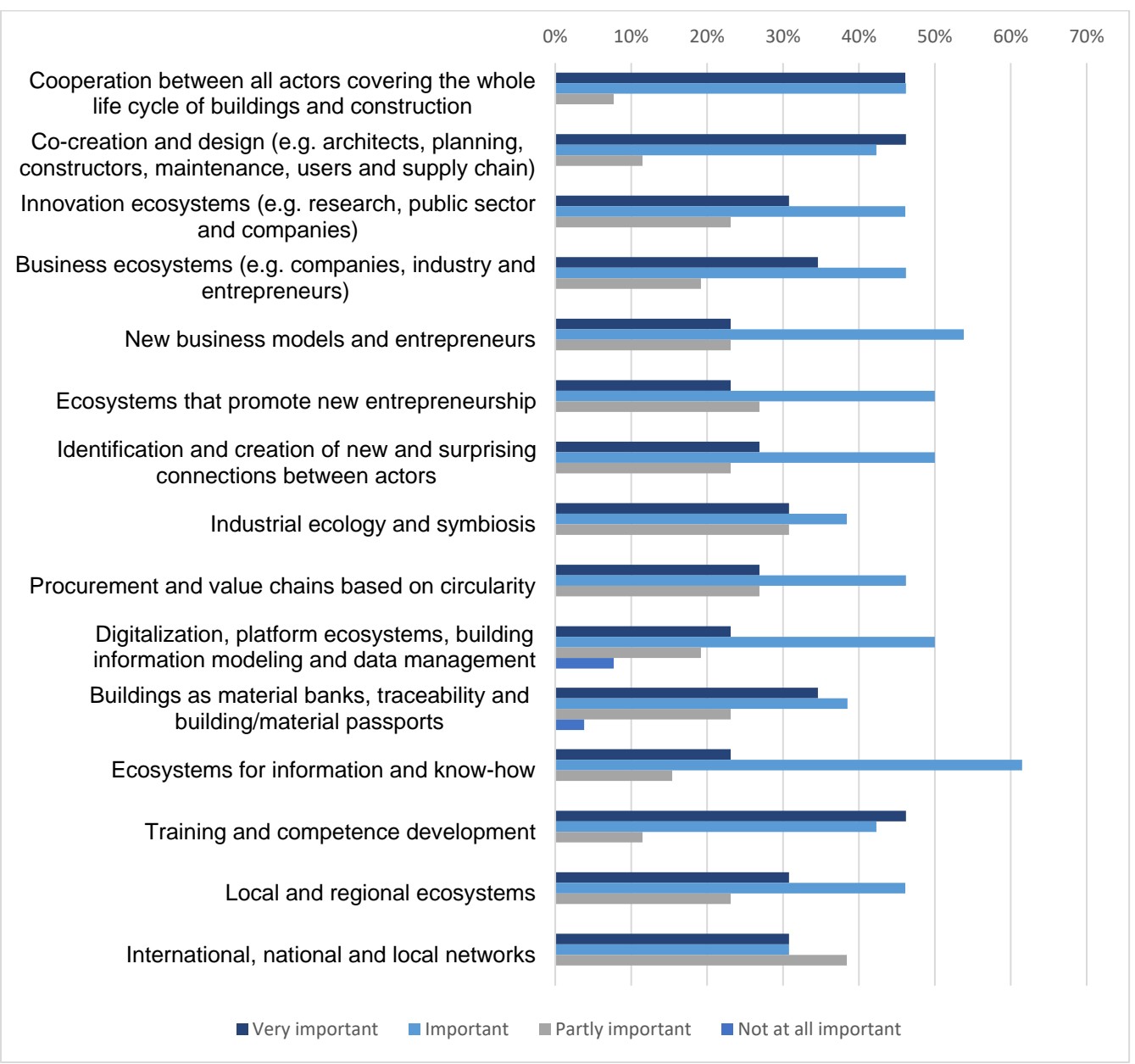

**Figure 12.** The importance of aspects related to CE ecosystems in the wood construction sector.

The promotion of CE in the building sector in the EU requires the promotion of new business models and reuse by policies; more coordinated actions and policies to advance circularity; and tools to advance circular supply chains and networks of operators [7]. In general, CE is a holistic concept, and silo structures can create barriers for associated development [18]. In addition, CE requires systems thinking, and the realization of a CE requires concerted action encompassing, e.g., research and innovation; policies; information and communication strategies; and new ways to measure societal well-being and wealth [18].

Our previous findings indicate that the most important steering approaches for a CE encompass (1) cooperation between the parties of the construction life cycle; (2) sustainability criteria for the whole life cycle (environmental, economic and social); (3) the loosening of waste and chemical legislation to promote product recycling and reuse; and (4) CE criteria for all project stages [65]. The enablers for business ecosystem development and deeper collaboration include trust building and communication among new ecosystem partners,

whereas barriers comprise a lack of clear and shared goals between various actors and insufficient end-user involvement [112].

It has been noted that sustainability transition in the construction industry is slowed down by limited ecosystem-level awareness of the benefits of using renewable construction materials [112]. There is also a lack of application of circular networks among operators, and important CE focus areas at the building level include, e.g., businesses for the networking of operators [7]. Socio-cultural changes are also a major CE challenge, more so than technological innovation [114]. In addition, CE transition and strategies to promote higher levels of circularity require socio-institutional changes in the whole product chain [114]. Business ecosystem thinking can be applied to multi-story wooden building projects to increase, e.g., awareness of sustainability issues; mutual learning; new insights based on research and development; reference values; and financial and employment benefits [115].

There is a need for new legislation and policies that create drivers and overcome barriers including stakeholder engagement and more participatory decision-making processes [7]. In addition, the barriers for the circular buildings sector and the circular design of buildings encompass the conservativeness of and the dependency throughout the building industry and the lack of political priority [56]. There are no industry-wide CE practices in firms in the construction industry [11]. However, there are promising developments in individual firms or supply chains (e.g., the purchase of CE materials and design using non-virgin materials) [11].

The main barriers to CE practices in the construction industry encompass the lack of collaboration, cooperation and knowledge transfer between actors; guidance from policy makers (e.g., guiding principles for public procurement); monitoring procedures (e.g., measurements and indicators); fragmentation; a lack of transparency; and isolation and the limited impact of CE projects [11]. There is a lack of CE knowledge and standard practices inside the construction industry, and there is a potential to create a Community of Practice in this field [12]. A CE of timber is not often addressed or implemented, and there is a need for (1) the policy- and regulation-driven promotion of the CE of structural timber; (2) the identification of actors in the whole value chain that can guarantee the economic value of waste materials [72].

An analysis of current networks from the business ecosystem perspective indicates that their full potential is not utilized; main actors need to develop their leadership skills, including the communication of shared goals and feedback; stronger end-user involvement is needed based on continuous communication (not just single projects); and a more inclusive approach to new business ecosystem participants is needed covering all phases such as planning, building, living and use [112]. The development of circular buildings requires (1) the co-creation of an ambitious vision including the involvement of stakeholders with relevant knowledge to promote supply chain collaboration and (2) new types of supply chain collaborations (a dynamic network covering all partners such as suppliers, designers and demolition/waste companies) with a focus on a new process design that integrates multiple disciplines (e.g., multidisciplinary problem solving) and the extension of responsibilities to actors along the whole building supply chain [86].

### 3.11. The Importance of CE Business Models and Associated Aspects in the Wood Construction Sector

The results indicate that the design for a CE, sustainability and long life cycles; the circularity of raw materials; the extension of product life cycles; renewable raw materials; and market creation for recovered and recycled materials, components and products were considered to be very important (Figure 13). Many other aspects were also considered to be important such as remanufacturing, reuse and conversion to new use of products, components and materials; recovery and recycling of resources; market creation for recovered and recycled materials, components and products; new ways to design products and services; digitalization and digital solutions and market places; and recovery and recycling of products, components and materials.

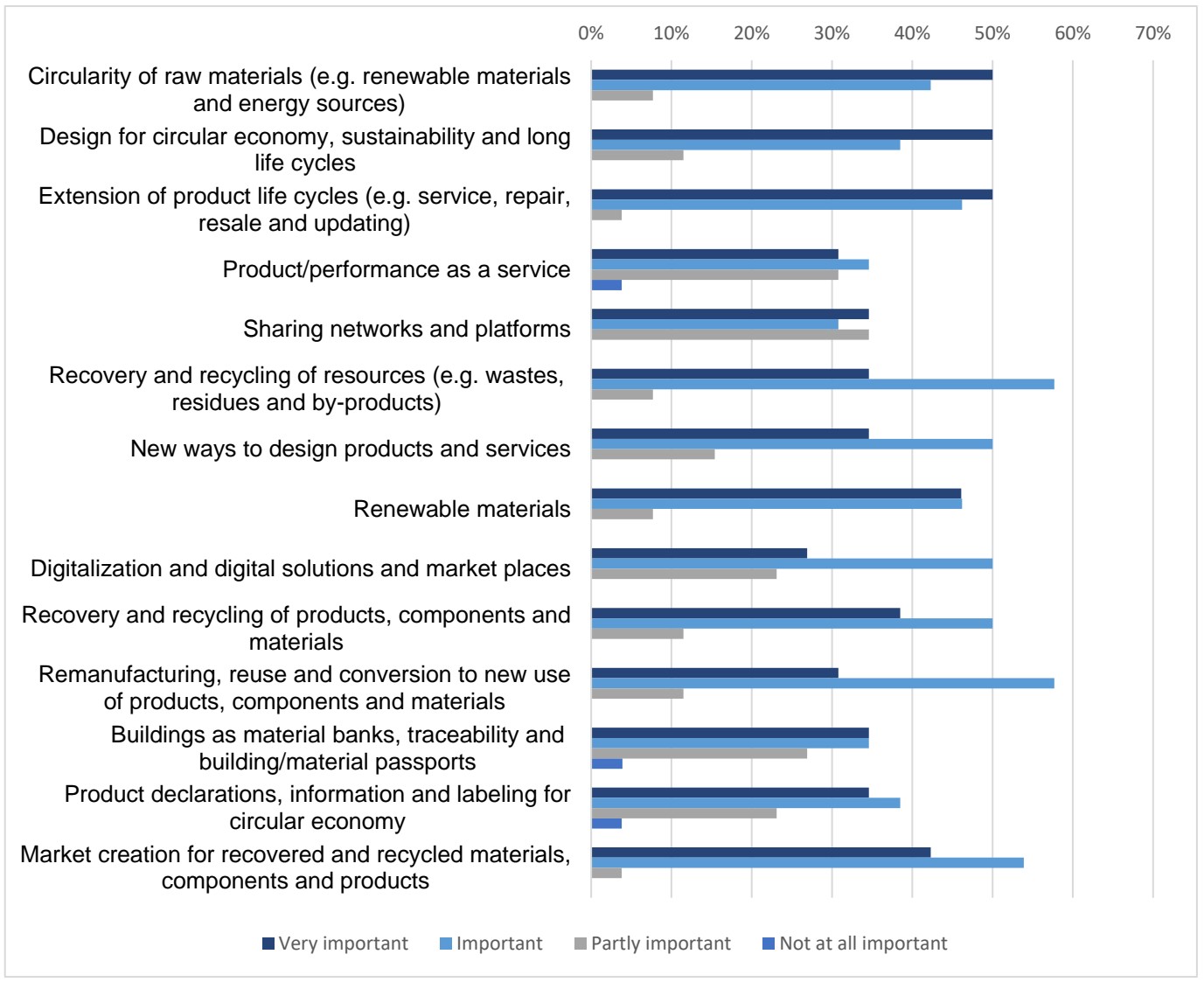

**Figure 13.** The importance of CE business models and associated aspects in the wood construction sector.

Our previous findings indicate that (1) the most important CE drivers and opportunities encompass products that are designed to be repairable and reusable and the sustainability and long-life cycles of products, components and materials [65]; (2) the sustainability of products needs to be considered early in the design phase considering life cycle thinking [66]; (3) the most important barriers to a CE encompass a low economic benefit associated with sorting and recycling (profitability); and insufficient monitoring data about construction wastes (e.g., quantity and quality) [65]; and (4) businesses can advance CE, e.g., through the use of renewable raw materials and new innovations and business models [66].

CE is linked to the societal trend towards intelligent decentralization, including new business models, and CE business models are typically based on reuse and extended service life via repair, remanufacturing, upgrades and retrofits and the recycling of materials to provide new resources [18]. In addition, all people play a major role in a CE, including stewardship, consumers as creators and users and skilled local workers [18]. The application of CE practices and business models contributes to the achievement of the UN SDGs, and there is a need for enhanced skills, training, capacity building and multistakeholder partnerships [116].

Previous studies have recognized the following: (1) the current CE practices in industry mainly focus on reusing construction and demolition waste, and there is a lack of clarity and insight into circular business models in this sector [14]; (2) companies can build on successful circular business models and create a CE ecosystem that encompasses customers, key partners and suppliers [15]; (3) a CE requires an understanding of demand, resource requirements and supply specifications [15]; (4) supply-side CE focus areas include the design of components for reuse, the design of products that can be upgraded and refurbished, circular supply logic, renewable energy, biomaterials and biochemicals, the use of recovered secondary materials and low-cost end-of-life recycling [15]; (5) a CE can promote sustainable innovations that focus on the social elements of a CE, collaboration among diverse actors, cultural practices and social engagements and practices [117].

In addition, companies can base their circular business models on, e.g., product life extension; circular supply chains; recovery and recycling; sharing platforms; and products as services [15]. Demand-side CE focus areas include ways to engage customers; the role of customers during and after product use; product development; and the evolution of resource requirements [15]. Sustainable innovation including the optimization of environmental, social and economic benefits of innovations contribute to the creation of lasting value for customers, employees, investors and the whole society [62]. In addition, sustainable choices need to be promoted by a framework of clear regulations, rules and standards with a focus on preventing the market access of products, systems and services that have a negative impact on society and the planet [62]. There is also a lack of application of circular business models, and the sustainability of circular strategies is not usually assessed (e.g., based on LCA) [7].

In general, the development of circular buildings requires new business and ownership models [86]. For example, businesses can use CE value retention options (e.g., 10Rs) to identify their possibilities to engage in CE, including the development of business models and the consideration of these value retention options in the context of both the product concept and design life cycle and the product produce and use life cycle [118]. In addition, the transformation from a linear economy to a CE requires the integration of business models and design strategies that include approaches, methods and tools to support CE [26]. The development and application of circular business models can support a transition towards sustainable development that includes the role of circular business model innovation aligned with the CE paradigm [119].

### 3.12. The Importance of Approaches to Assess CE Performance in the Wood Construction Sector

The results indicate that the use of renewable energy; building information modeling; forest and wood product certification; and criteria and indicators for sustainable forest management were considered to be very important by the respondents (Figure 14). In addition, data management and monitoring; CE management, assessment and reporting; created new innovations, business models, ecosystems and design approaches; the use of renewable raw materials; and product declarations, information and labeling for CE were considered to be important by many of the respondents.

Our previous findings indicate that the very important aspects of CE encompass the following: (1) the maintenance of existing buildings; (2) sustainability and long life cycles of products, components and materials; (3) cooperation between the parties of the life cycle of construction; (4) products that are designed to be repaired and reused; and (5) sustainability criteria for the life cycle of construction (environmental, economic and social aspects) [65]. Currently, environmental assessments of buildings are typically based on reactive methods that assess building designs after their completion, and LCA is applied late in the design process, which does not allow significant influence on the design or a large impact on the environmental performance of buildings [64].

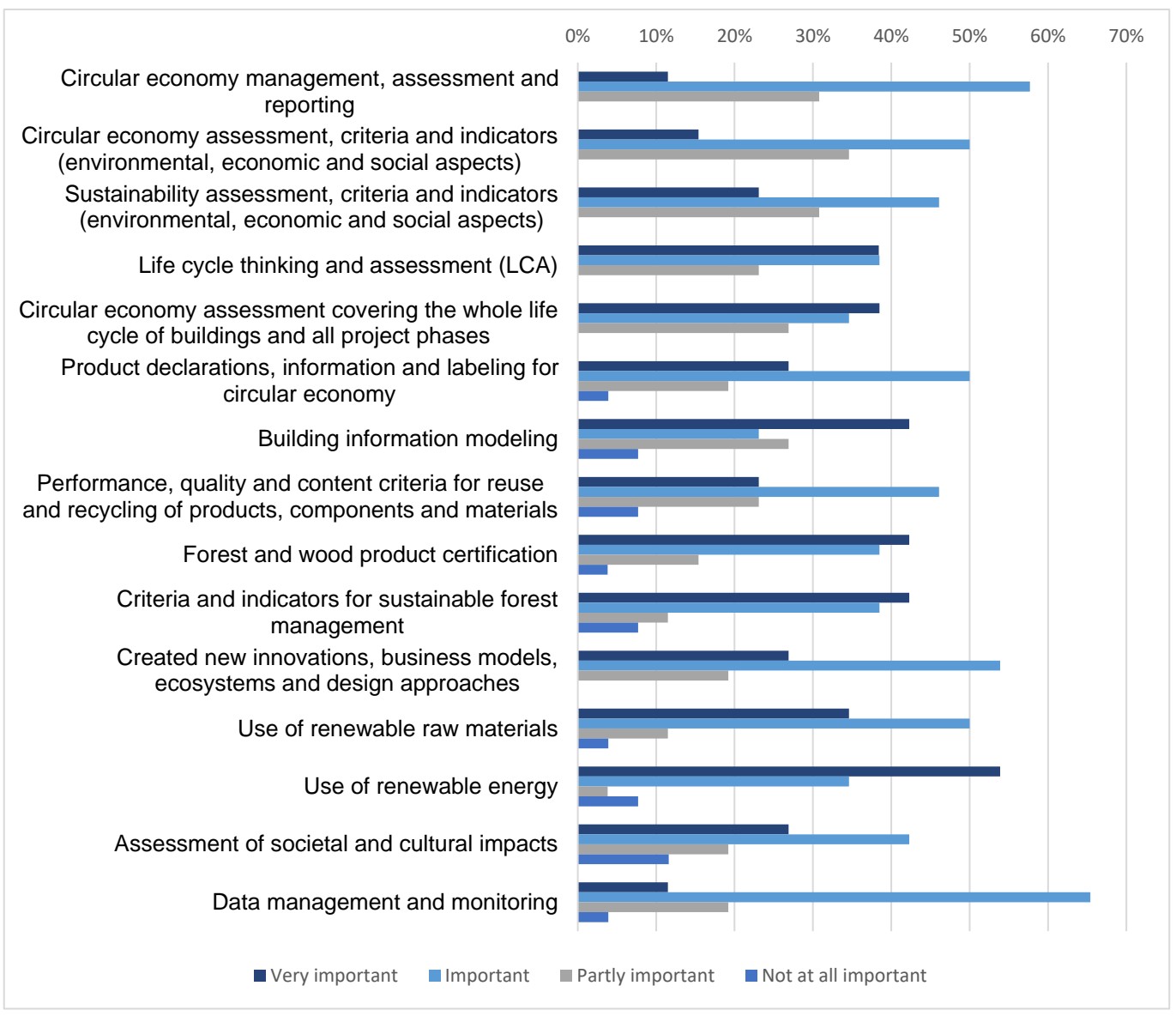

**Figure 14.** The importance of approaches to assess CE performance in the wood construction sector.

In general, CE transition and circularity need to be supported by appropriate sustainability assessment tools [4]. Construction industry actors should be proactive in creating environmental, social and economic sustainability indicators (e.g., covering design, construction, operations and dismantling) to promote building sector sustainability and sustainable construction practices in both developed and developing countries [120]. The consideration and selection of sustainable construction materials requires a focus on all environmental impacts; social and economic aspects; the finite nature of the resource; a reduction in negative impacts on both people and the environment; and the promotion of social well-being and economic viability [82].

Previous studies have recognized that (1) a CE of timber is not often addressed or implemented, and there is a need for the development of assessment criteria for the reuse of structural timber [72]; (2) a broader approach to sustainability encompassing the integration of sustainability assessment, CE and life cycle thinking and comprehensive sustainability strategies covering the whole building life cycle are still new in the construction industry [121]; (3) there is no consensus on how to measure progress of the CE transition process including its effects on circularity, the environment and the economy [114]; (4) a cleaner building industry can be promoted by the integration of sustainability and CE assessments

and decision-making frameworks [8]; and (5) the environmental performance of wood is dependent on its use and applied environmental indicators [74].

In addition, it is essential to keep in mind in the context of design choices that a focus on a single environmental indicator (e.g., the carbon footprint) does not cover the whole spectrum of sustainability or all other environmental impacts [71]. Therefore, public authorities should balance simplified approaches and complete environmental approaches that require the systemic control of complex environmental impacts [71]. In addition, life cycle thinking is needed to ensure that CE provides environmental benefits by extending the life of products and services and minimizing environmental burdens [122].

It has been noted that there is significant potential for environmental impacts reductions (e.g., global greenhouse gas emissions) associated with buildings, and the creation of a net-zero carbon built environment requires life cycle thinking and a reduction in whole-life impacts of buildings [64]. In addition, the sustainability of wood construction requires a focus on associated assessment and rating systems and approaches, and there is a need for local LCA approaches that reflect the recognized environmental benefits of using wood in construction [123]. For example, proactive methods provide means to reduce the whole-life impacts of buildings, and they can help to guide decisions in the early design stages and to introduce LCA knowledge to the design process to influence the environmental impacts of buildings [64].

In addition, the promotion of environmental performance and the sustainability assessment of buildings requires a focus on (1) all aspects of sustainability; (2) the suitability of the assessment criteria and indicators; (3) the maintenance, modernization and renovation of existing buildings; (4) easier deconstruction and recycling methods of both renovated and new buildings in the design stage; (5) current and future environmental impacts; and (6) legislative initiatives and requirements that consider the conservation of natural resources [124]. Important guidelines and tools include, e.g., end-of-waste criteria; pre-demolition audits; traceability guidelines; and material passports [7].

### 3.13. The Use or Introduction of Approaches to Promote CE of Wood Construction

The results indicate that the maintenance, renovation, refurbishment, retrofitting and conversion of existing buildings; co-creation and design; training and competence development; the cascading use of wood; and new and innovative design for CE were in use by many of the respondents (Figure 15). Many approaches such as CE management, assessment and reporting; new and innovative design for CE; cooperation between all actors covering the whole life cycle of buildings and construction; and the cascading use of wood were in consideration.

Training and competence development; CE, innovation and business ecosystems; demolition plans and auditing; market creation for recovered and recycled materials, components and products; digitalization and data management and monitoring systems; performance, quality and content criteria for recycled and reused materials, components and products; and buildings as material banks, traceability and building/material passports were coming into use according to many of the respondents. In addition, some respondents considered that, for example, demolition plans and auditing and market creation for recovered and recycled materials, components and products were not in use or coming into use.

Previous studies have recognized that (1) sustainability covering environmental, social and economic aspects in the construction industry requires the collaboration of all supply chain actors with a particular focus on the design phase and on the alignment of the design process and the supply chain to enhance sustainability performance [125]; (2) positive environmental attributes are often associated with wooden multi-story buildings, and further research is needed on environmental, social and economic impacts of multistory buildings including their influence on the implementation of associated projects nationally [126]; (3) wood attributes and quality aspects (e.g., sustainability, social, economic and technical properties) affect the perceived quality of wooden building materials, and con-

sumer behavior related to wooden building materials is influenced by various personal and situational variables [127]; (4) multi-story wooden buildings provide options to promote sustainable development [128]; and (5) the main motivations for using wood in multi-story and nonresidential construction projects encompass, e.g., sustainability, technical aspects, the aesthetics of wooden structures, rapid construction process and costs, whereas the barriers include, e.g., the culture of the industry, the lack of expertise and the failure to implement building codes [129].

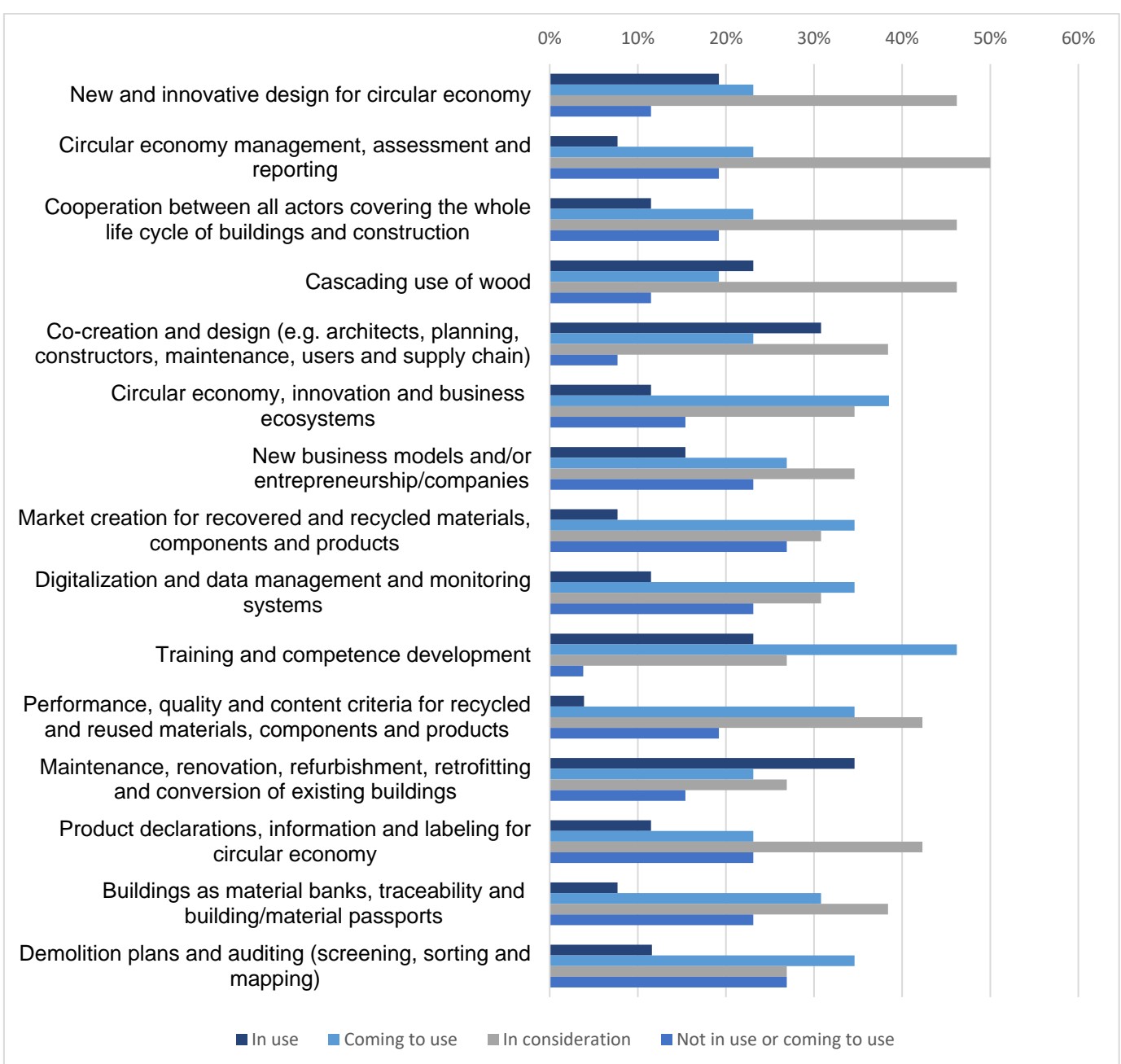

**Figure 15.** The use or introduction of approaches to promote CE of wood construction.

In general, the number of timber buildings in less traditional urban applications is increasing due to sustainability merits [130]. In addition, there is increasing support for the use of wood in multi-story construction, and it creates benefits such as the support of local industries by locally sourcing renewable building materials [131]. Most studies confirm that wooden materials generally have lower embodied energy compared to traditionally

used construction materials, and the main advantages related to lower environmental impacts of wooden buildings are related to wooden design in the construction phase and low impacts in the end-of-life life cycle stage due to full recyclability [83]. However, neither end-users or companies fully recognize sustainability aspects associated with wooden multi-story construction [112]. There is also a need to communicate sustainability and the physical properties of wood to consumers by connecting them to meaningful topics such as local production, the nostalgic aspects of wooden materials and the pleasant ambiance of wooden living [130].

Previous studies have also identified many essential aspects including the following: (1) the achievement of a higher share of wooden building material in construction and the fact that retrofitting requires a mix of policy instruments that are adapted to the specific national cultural, political and economic features particularly in terms of wood materials [77]; (2) wood construction can be promoted via prefabrication, clearer business concepts, education and training (e.g., design), information on the environmental performance of wood as a building material and support for and collaboration among architects and engineers [93]; (3) the barriers to the use of wood in multi-story construction encompass the limited distribution of information, inefficient policy measures and the limited number of industry actors [131].

In addition, there is a need for (1) the promotion and better acknowledgement of the broader sustainability impacts and positive health impacts of wood in the whole society [79]; (2) a deeper understanding of the sustainability issues that affect the acceptability of wood among various types of consumers [128]; and (3) knowledge generation to support a circular bioeconomy that includes wood-frame multi-story constructions across the innovative system; interest from large building companies; low-carbon public procurement criteria; new municipal organizational practices (actor–networks); and the development of guidelines for best practices [132].

## 4. Conclusions

The findings of this study suggest that sectoral companies consider it to be important to integrate various CE approaches, concepts and aspects into the design and construction of wood buildings that covers their whole life cycles, all phases of construction and all actors in the overall supply chain. Important focus areas encompass, for example, sustainability and long life cycles of products, components and materials; co-creation and cooperation covering the whole life cycle of construction and building and the whole supply chain; training and competence development; a reduction in the use of chemicals and dangerous substances; and buildings as material banks, traceability and building/material passports.

The limitations of this study encompass the fact that the results are based on the views of the company respondents and do not fully represent the overall views on development in this sector as perceived by industry and companies as a whole. However, they provide a good indication of what are the important focus areas in this particular sector.

On a broader level, research and development work is needed on, for example, the introduction of CE business models and market creation; the integration of CE principles and approaches into company practices; tools to manage and assess all CE aspects; the establishment of supporting and enabling ecosystems; systems and tools for the full utilization of information and data; the type of new entrepreneurs needed to promote CE development; and holistic, new and innovative approaches to co-creation and design. In addition, all aspects related to wood are important, such as sustainable forest management, forest and wood product certification and overall cascading use encompassing the full utilization of the full potential of wood resources through sequential and multiple uses aiming at the maintenance of the highest possible utility, usability and value at all times and uses.

The following conclusions can be highlighted based on the findings: (1) companies perceive CE as an important concept, and they are increasingly interested in and applying CE approaches; (2) many companies are already using approaches to advance CE, including

focusing on the sustainability and long life cycles of products, components and materials as well as on the maintenance, renovation, refurbishment, retrofitting and conversion of existing buildings; (3) designs for new CEs, sustainability, innovation and business ecosystems will be applied by many companies in the near future; (4) specific tools such as product certification and building information modeling are very important; (5) specific aspects such as designs for CEs, sustainability, and the long life cycles of products and the extension of product life cycles are very important; (6) the building of ecosystems requires strong cooperation between all actors covering the whole life cycle of buildings and construction; (7) actions such as the reduction in the use of chemicals and dangerous substances are required to both promote the cascading use of wood and advance the integration of CE into this sector; (8) very important CE aspects in this sector encompass co-creation and cooperation across the whole life cycle of construction and building, including the whole supply chain, in addition to training and competence development; (9) communication between designer and both client and constructor is very important; (10) more companies need to be familiarized with essential CE aspects such as the cascading use of wood, products as services, sharing platforms and the assessment and measurement of CE.

It is noteworthy that many essential elements of CE were in use, coming into use or in consideration by many of the sectoral companies. These findings and identified important approaches, concepts and aspects can potentially contribute to further the development of CE in the wood construction sector via their appropriate integration into overall building design and construction, covering all actors based on a full system level and life cycle perspective. Essential future steps for companies could encompass, for example, the consideration of the whole life cycle of products, components and materials from forests to circularity with the cascading use of wood. In addition, companies could introduce and apply CE management and assessment approaches to fully integrate these aspects into their operations. There are also major benefits for companies associated with co-creation and ecosystems covering the whole life cycle of construction and buildings, taking into account the importance of the design phase. Future research should focus on the further assessment of CE development in this sector, covering the identified important focus areas and their adoption by companies.

**Author Contributions:** Conceptualization, R.H. and D.S.; methodology, R.H. and D.S.; software, R.H.; validation, R.H.; formal analysis, R.H.; investigation, R.H.; resources, R.H.; data curation, R.H.; writing—original draft preparation, R.H.; writing—review and editing, R.H. and D.S.; visualization, R.H. and D.S.; supervision, R.H.; project administration, R.H. All authors have read and agreed to the published version of the manuscript.

**Funding:** This research received no external funding.

**Institutional Review Board Statement:** The study was conducted according to the guidelines of the Declaration of Helsinki and ethical review process guidance at the Aalto University was followed.

**Informed Consent Statement:** Not applicable.

**Data Availability Statement:** Research data are available from the authors upon request.

**Conflicts of Interest:** The authors declare no conflict of interest.

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
