# Peer review of "Circular Economy Development in the Wood Construction Sector in Finland"

_sustainability, doi:10.3390/su15107871_

Round 1

Reviewer 1 Report

This study aimed at exploring, identifying, analyzing and synthesizing the current state of and future outlook on CE development in the wood construction sector in Finland as perceived by various sectoral companies. The paper has certain research value, but it is important to take note of several issues.

1. Abstracts should not be directly listed in a series of research content, which would make the paper appear less connected and logical. At the same time, it is not recommended to break up the abstract section. 

2. Can the number of research samples be increased?

3. Is there a better way to present the relationship after data analysis in the article? 

4. Most of the text in the following parts of the article is a direct representation of the data, without logical analysis. It is not recommended to express this as it would increase workload.

5. The data source and method for the entire article are based on document qualitative analysis and questionnaire survey. Can this sample data accurately reflect the overall development of the Finnish timber and construction industry? However, this qualitative data may not necessarily reflect the development trend of the industry in a precise manner.

6. Overall, the overall reading experience of the entire article appears to be industry report rather than independent research article.

Minor editing of English language required

Author Response

This study aimed at exploring, identifying, analyzing and synthesizing the current state of and future outlook on CE development in the wood construction sector in Finland as perceived by various sectoral companies. The paper has certain research value, but it is important to take note of several issues.

  1. Abstracts should not be directly listed in a series of research content, which would make the paper appear less connected and logical. At the same time, it is not recommended to break up the abstract section. 

Reply:

Thank you for this comment. This has been addressed and the abstract is now revised accordingly. 

However, We do consider it important to present the main parts of an original research article in the abstract.

  1. Can the number of research samples be increased?

Reply: Unfortunately not and authors did their best to make the response rate higher (e.g. reminders and new emails). The surprisingly low response rate was a surprise and probably due to the novelty of the CE concept for the respondents and the characteristics of the survey which was broad and detailed including many challenging questions about development and current status in companies as well as probably many new concepts as well. Many started the survey and did no finalize. This survey was more ambitious and challenging than any of our previous ones.

However, we consider that the current rate and obtained results provide a good indication of views in this particular sector. Furthermore, they lay a basis for further work and discussion. Interestingly, the low rate may also suggest that it is very timely and important to address this issue in this sector as at the policy level (EU and national) very much is going on and it is interesting and relevant to know what is actually going on in companies and what they think about development.

  1. Is there a better way to present the relationship after data analysis in the article? 

Reply:

Thank you for this comment. We think the present style is good and logical visually. This style was adopted after peer-review of our previous article “Circular Economy Development in the Construction Sector in Japan” and it was considered to be informative.

  1. Most of the text in the following parts of the article is a direct representation of the data, without logical analysis. It is not recommended to express this as it would increase workload.

Reply:

Thank you for this comment. However, we do consider that results need to be addressed in the text in each section (in a brief manner). As this is a qualitative study this is highly relevant as results are not just data but information. This style was adopted in our previous article “Circular Economy Development in the Construction Sector in Japan” and it was considered to be informative.

  1. The data source and method for the entire article are based on document qualitative analysis and questionnaire survey. Can this sample data accurately reflect the overall development of the Finnish timber and construction industry? However, this qualitative data may not necessarily reflect the development trend of the industry in a precise manner.

Reply:

Thank you for this comment. The article reflects the current state of and future outlook on CE development in the wood construction sector in Finland as perceived by various sectoral companies. These companies represent both architectural and construction sector companies. We fully understand the focus and limitations of the study. We have added a sentence about the limitations in conclusions about this. We are also open to a revision of the title if journal editors support that idea and it can be improved.

  1. Overall, the overall reading experience of the entire article appears to be industry report rather than independent research article.

Reply:

Thank you for this comment. However, we do not fully understand this comment as the article is based on qualitative research method and discussion with 133 references. In addition, we have applied this approach in multiple previous scientific articles.

Reviewer 2 Report

Dear Authors,

the paper applied a qualitative research approach and a questionnaire survey as the specific method. The aim of this study is to fill a gap in research and contributes to better understanding of the current state of and future outlook on CE development in the wood construction sector. The topic is interesting and deserves attention. 

The introduction is interesting and shows the different points of view of the circular economy. However, it may also be interesting to introduce the concept of "ecodesign" to increase the field of research. How the ecodesign approach can be help the CE development? Are there any examples already present in the scientific literature? What are the limits? what are the problems? For example these researches are interesting to extend the introduction. 10.1016/j.jclepro.2022.130866; 10.1007/s12008-023-01249-0

How were the questions written? Was any logic followed?How were the questionnaires sent? Where were they made?

Why were those questions chosen?

Please detail your conclusions to provide more information. it is necessary to summarize the results of the survey

The paper is well write and well structured

Author Response

Dear Authors,

the paper applied a qualitative research approach and a questionnaire survey as the specific method. The aim of this study is to fill a gap in research and contributes to better understanding of the current state of and future outlook on CE development in the wood construction sector. The topic is interesting and deserves attention.

The introduction is interesting and shows the different points of view of the circular economy. However, it may also be interesting to introduce the concept of "ecodesign" to increase the field of research. How the ecodesign approach can be help the CE development? Are there any examples already present in the scientific literature? What are the limits? what are the problems? For example these researches are interesting to extend the introduction. 10.1016/j.jclepro.2022.130866; 10.1007/s12008-023-01249-0

Reply:

Thank you for this highly relevant comment and supportive suggestion for addition. We have now included new references in the introduction section about “ecodesign” and “product design”encompassing these articles:

[22]Marconi, M.; Favi, C. Eco-design teaching initiative within a manufacturing company based on LCA analysis of company product portfolio. Journal of Cleaner Production, 2020, 242, 118424. https://doi.org/10.1016/j.jclepro.2019.118424

[23]Spreafico, C.; Landi, D. Using Product Design Strategies to Implement Circular Economy: Differences between Students and Professional Designers. Sustainability 2022, 14, 1122. https://doi.org/10.3390/su14031122

[24]Ipsen, K.L.; Pizzol, M.; Birkved, M.; Amor, B. How Lack of Knowledge and Tools Hinders the Eco-Design of Buildings—A Systematic Review. Urban Sci. 2021, 5, 20. https://doi.org/10.3390/urbansci5010020

[25]Spreafico, C.; Landi, D. Investigating students’ eco-misperceptions in applying eco-design methods. Journal of Cleaner Production, 2022, 342, 130866, https://doi.org/10.1016/j.jclepro.2022.130866.

[26] ​​ Bocken, N​.​ M., Pauw, P.​ ​I​. ​de., Bakker ​ ​C​.​  â€‹and Grinten., ​B​.​ van der​, : Product design and business model strategies for a circular economy​, ​​pp.​308-320​, ​Journal of Industrial and Production Engineering​​, 2016, ​Vol​.​33,Issue 5​, ​ https://doi.org/10.1080/21681015.2016.1172124

[27] ​​Hollander​., M, den​, Bakker​., ​C.A. â€‹and ​Hultink​., ​Erik Jan: Product Design in a Circular Economy: Development of a Typology of Key Concepts and Terms: Key Concepts and Terms for Circular Product Design, Journal of Industrial Ecology,2017, 21,3, https://doi.org/10.1111/jiec.12610

How were the questions written? Was any logic followed?How were the questionnaires sent? Where were they made?

Reply:

Thank you for this comment.

The questions were written as “how important is..”; “how familiar are you with..”;”how important are..”; or “are the following in use or will you introduce them..”. A sentence about this was added.

The survey themes encompassed (1) the importance of the CE concept as a part of building design and construction; (2) the familiarity of CE aspects in the construction sector; (3) the importance of the main principles of CE in the built environment; (4) the importance CE aspects in the design of wooden buildings; (5) the importance of CE aspects related to wood materials, components and products; (6) the importance of CE aspects in the wood construction sector; (7) the importance of approaches to integrate CE into wood construction; (8) the use or introduction of approaches to promote CE in the design of wooden buildings; (9) the importance of aspects related to cascading use of wood for wood construction; (10) the importance of aspects related CE ecosystems in the wood construction sector; (11) the importance of CE business models and associated aspects in the wood construction sector; (12) the importance of approaches to assess CE performance in the wood construction sector; and (13) the use or introduction of approaches to promote CE of wood construction.

Why were those questions chosen?

Reply:

The survey themes, questions and answering options were based on (1) literature re-view and previous studies; (2) our previous studies; and (3) assessment of the global, EU and national operational environment for CE development in the construction sector with special emphasis on wood construction. In addition, face validity (peer review) [48] was applied to check the quality of the whole questionnaire.

How were the questionnaires sent? Where were they made?

Reply:

The survey was anonymous and voluntary and its respondents included architects and company representatives (e.g. managers, project managers, designers and experts) in the construction sector.

The survey was sent directly to respondents via email. “via email” was added.

All the questions and the questionnaire was made by the authors and the applied survey system was the Webropol. A sentence addressing this was added.

Please detail your conclusions to provide more information. it is necessary to summarize the results of the survey

Reply:

This has been addressed and there is a new section about this in conclusions.

Reviewer 3 Report

The overall aim of this study was to explore, identify, analyze and synthesize the current state of and future outlook on CE development in the wood construction sector in Finland as perceived by various sectoral companies.

This study applied a qualitative research approach and the chosen specific method was a questionnaire survey which was applied as an online survey that was sent via email directly to respondents.

The findings of this study suggest that sectoral companies consider it to be important to integrate various CE approaches, concepts and aspects into design and construction of wood buildings covering their whole life cycles, all phases of construction and all actors in the overall supply chain. 

There are still some problems that need to be paid attention to and improved, such as:

(1) Why choose 13 topics from the article? The relevance of the selected research topic to the main research content of the paper should be properly elaborated. 

(2) It should be explained in more detail which aspects should be focused on in the future of the study "CE development of Finnish timber structure building industry". 

(3) More detailed analysis should be made on the results of the survey data to analyze the causes of this phenomenon.

(4) The document marking error "[6-14]" appears on page P2, and the whole text should be checked and marked uniformly. 

(5) Some pages of the article have too much blank space, which should be appropriately deleted, such as P6, P8, P10, P14, etc.

(6) The title of Figure 1 and Figure 2 is suggested to be in the center, which is consistent with the full text. 

The overall aim of this study was to explore, identify, analyze and synthesize the current state of and future outlook on CE development in the wood construction sector in Finland as perceived by various sectoral companies.

This study applied a qualitative research approach and the chosen specific method was a questionnaire survey which was applied as an online survey that was sent via email directly to respondents.

The findings of this study suggest that sectoral companies consider it to be important to integrate various CE approaches, concepts and aspects into design and construction of wood buildings covering their whole life cycles, all phases of construction and all actors in the overall supply chain. 

There are still some problems that need to be paid attention to and improved, such as:

(1) Why choose 13 topics from the article? The relevance of the selected research topic to the main research content of the paper should be properly elaborated.     (2) It should be explained in more detail which aspects should be focused on in the future of the study "CE development of Finnish timber structure building industry".     (3) More detailed analysis should be made on the results of the survey data to analyze the causes of this phenomenon.    (4) The document marking error "[6-14]" appears on page P2, and the whole text should be checked and marked uniformly.     (5) Some pages of the article have too much blank space, which should be appropriately deleted, such as P6, P8, P10, P14, etc.     (6) The title of Figure 1 and Figure 2 is suggested to be in the center, which is consistent with the full text. 

Author Response

The overall aim of this study was to explore, identify, analyze and synthesize the current state of and future outlook on CE development in the wood construction sector in Finland as perceived by various sectoral companies.

This study applied a qualitative research approach and the chosen specific method was a questionnaire survey which was applied as an online survey that was sent via email directly to respondents.

The findings of this study suggest that sectoral companies consider it to be important to integrate various CE approaches, concepts and aspects into design and construction of wood buildings covering their whole life cycles, all phases of construction and all actors in the overall supply chain.

There are still some problems that need to be paid attention to and improved, such as:

(1) Why choose 13 topics from the article? The relevance of the selected research topic to the main research content of the paper should be properly elaborated.

Reply:

We have added a new sentences on this to materials and methods section. The whole relevant section is now like this:

The overall approach was to explore CE development in this sector through themes and issues raised in previous studies; literature; and various documents in this particular field of study such as strategical and policy initiatives. In addition, our own previous research related to both CE and the construction sector in particular was duly considered in the formulation of the themes and questions. In brief, the survey themes, questions and answering options were based on (1) literature review and previous studies; (2) our previous studies; and (3) assessment of the global, EU and national operational environment for CE development in the construction sector with special emphasis on wood construction. In addition, face validity (peer review) [482] was applied to check the quality of the whole questionnaire.

(2) It should be explained in more detail which aspects should be focused on in the future of the study "CE development of Finnish timber structure building industry".

Reply:

Thank you for this valid comment and suggestion. We have added these sentences to the conclusions section:

Essential future steps for companies could encompass, for example, consideration of the whole life cycle of products, components and materials from forests to circularity covering cascading use of wood. In addition, companies could introduce and apply CE management and assessment approaches to fully integrate these aspects into their operations. There are also major benefits for companies associated with co-creation and ecosystems covering the whole life cycle of construction and buildings taking into account the importance of the design phase.

(3) More detailed analysis should be made on the results of the survey data to analyze the causes of this phenomenon.

Reply:

We have added these:

Results

3.1

In Finland, CE has been an important focus area of some major societal actors including international events and specific initiatives that focus on the construction sector. Thus, the perceived importance of the concept may have been promoted through these continuous efforts.   

3.2

Similarly to results in the section 3.1, the continuous efforts to create awareness of and to advance CE in Finland including sectoral initiatives have enhanced the familiarity of companies with the concept and associated key aspects.    

3.3

It may partly explain the importance of energy and waste aspects that they are already familiar from various contexts such as resource and energy efficiency compared to the relative new CE concept.   

3.8

Interestingly, many respondents highlight long-term sustainability with particular emphasis on life cycle thinkingencompassing consideration of all phases from materials to renovation and retrofitting.

3.9

Interestingly, many respondents seem to familiar with practical challenges related to the use of chemicals and dangerous substances as it was perceived as a very important aspect to be addressed.

Regarding other results section we consider that the survey data speaks for itself and the discussion with references is sufficient to address this concern. This also true for all sections in addition to above additions. We have to be careful in qualitative research about making strong statement about respondents underlying reasons for giving particular answers.

(4) The document marking error "[6-14]" appears on page P2, and the whole text should be checked and marked uniformly.

Reply:

All reference numbers have been revised starting from 22.

Marking of [6-14) has been corrected to [6-14]

( = ]

That may have caused the error.

(5) Some pages of the article have too much blank space, which should be appropriately deleted, such as P6, P8, P10, P14, etc.

Reply:

This will be addressed jointly with the editors.

Large number of figures also affects spacing.

(6) The title of Figure 1 and Figure 2 is suggested to be in the center, which is consistent with the full text.

Reply:

Addressed and up to editorial advice on style.

Round 2

Reviewer 1 Report

The manuscript has been supplemented and modified by the authors, but I still think it is a literature review. It is recommended to select "Review" as the article type instead of "Article".

The author did not provide a valid response to the issues I raised in the article, so I still believe that further revisions are needed, at least in terms of the sample size and research methodology used in the article.

Minor editing of English language is required.

Author Response

The manuscript has been supplemented and modified by the authors, but I still think it is a literature review. It is recommended to select "Review" as the article type instead of "Article".

Reply:

Thank you for your further comments aimed at improving the manuscript. We consider that it cannot be a literature review because a qualitative research approach has been properly applied and it is not an industry report since it is research encompassing all scientific elements, approaches and parts as well as forms a scientific article (with aims, materials and methods, results and discussion and conclusions). It is a research article that includes extensive use of references as it should be particularly in discussion with the results sections. Companies were the target group for the study and respondents worked in those companies.

We have applied the same qualitative research approach [1] as in numerous previous original research articles. Moreover, we have applied questionnaire survey as a specific qualitative research approach [2-10].

References:

  1. Saldana, J. Fundamentals of Qualitative Research; Oxford University Press: Oxford, UK, 2011; pp. 3–30, ISBN-10:0199737959.
  2. Patten, M. L. Questionnaire research: a practical guide, 4th Ed.; Routledge: New Tork, USA, 2014. https://doi.org/10.4324/9781315265858
  3. Gillham, B. Developing a questionnaire, 2nd ed.; Continuum international publishing group: London, UK, 2007. ISBN-10:0826496318.
  4. Hirsjärvi, H.; Remes, P.; Sajavaara, P. Tutki ja kirjoita. 13th ed.; Tammi, Helsinki, 2007. ISBN 978-951-26-5635-6.
  5. Birmingham, P.; Wilkinson, D. Using Research Instruments: A Guide for Researchers, Taylor & Francis Group, 2003. ISBN 9780415272797.
  6. Fink, A. How to conduct surveys: a step-by-step guide, 4th ed.; Sage, 2009. ISBN-10: 141296668X.
  7. Oppenheim, A.N. Questionnaire design, interviewing and attitude measurement, Pinter: London, Washington, 1997. ISBN-10:0826451764.
  8. Saris, W.E.; Gallhofer, I.N. Design, Evaluation, and Analysis of Questionnaires for Survey Research, John Wiley & Sons, 2007. ISBN-10:0470114959.
  9. Sudman, S.; Bradbrun, N.M. Asking questions: a practical guide to questionnaire design. The Jossey-Bass series in social and behavioural sciences. Jossey-Bass Publishers, 1982. 9780985895464.
  10. Peterson, R. A. Constructing effective questionnaires, SAGE Publications: Thousand Oaks, CA, USA, 2000. doi: 10.4135/9781483349022

The author did not provide a valid response to the issues I raised in the article, so I still believe that further revisions are needed, at least in terms of the sample size and research methodology used in the article.

Thank you for your further comments aimed at improving the manuscript. Regarding methodology we have applied a well established qualitative research approach including a specific method, the questionnaire survey.

Regarding the sample size matter we consider it to be sufficient. We sent and resent the emails to respondents before submitting the manuscript but were not able to raise the number of respondents. This is most likely due to the challenging nature of both the topic (CE) and the survey. The respondents have various educational backgrounds and knowledge about CE and they may also consider company privacy ect in responding. Also many of them may simply not know what is being implemented or what is going to happen and are unsure about answering. Many started the survey and did not finnish and this also supports these possibilities. In addition, raising of the sample size can be asked for in all cases and it is possible in some cases but not all the cases. Moreover, the qualitative approach is commonly applied to topics like in this manuscript particularly when the theme of the research is new or unfamiliar in the field.

Thus, we do consider that the number of respondents is sufficient for a research article and this study clearly addresses a gap in research and there are no similar studies in Finland. This ambitious approach is also valid in terms of going beyond state-of-the-art in this specific line of research and all limitations are clearly expressed.

"The questionnaire was sent to respondents in architectural companies (150 in total) and in construction companies (150 in total) via email. All the respondents received the same survey and associated questions. The response rate was 8,7% (n. 26) covering 8 construction companies; 7 architectural companies; 6 manufacturing companies; 2 contractor companies; 1 company focused on projects; and 2 companies in other fields (Figure 1)."

Previous comments and associated replies from Round 1:

This study aimed at exploring, identifying, analyzing and synthesizing the current state of and future outlook on CE development in the wood construction sector in Finland as perceived by various sectoral companies. The paper has certain research value, but it is important to take note of several issues.

  1. Abstracts should not be directly listed in a series of research content, which would make the paper appear less connected and logical. At the same time, it is not recommended to break up the abstract section. 

Reply:

Thank you for this comment. This has been addressed and the abstract is now revised accordingly. However, We do consider it important to present the main parts of an original research article in the abstract.

  1. Can the number of research samples be increased?

Reply: Unfortunately not and authors did their best to make the response rate higher (e.g. reminders and new emails). The surprisingly low response rate was a surprise and probably due to the novelty of the CE concept for the respondents and the characteristics of the survey which was broad and detailed including many challenging questions about development and current status in companies as well as probably many new concepts as well. Many started the survey and did no finalize. This survey was more ambitious and challenging than any of our previous ones.

However, we consider that the current rate and obtained results provide a good indication of views in this particular sector. Furthermore, they lay a basis for further work and discussion. Interestingly, the low rate may also suggest that it is very timely and important to address this issue in this sector as at the policy level (EU and national) very much is going on and it is interesting and relevant to know what is actually going on in companies and what they think about development.

  1. Is there a better way to present the relationship after data analysis in the article? 

Reply:

Thank you for this comment. We think the present style is good and logical visually. This style was adopted after peer-review of our previous article “Circular Economy Development in the Construction Sector in Japan” and it was considered to be informative.

  1. Most of the text in the following parts of the article is a direct representation of the data, without logical analysis. It is not recommended to express this as it would increase workload.

Reply:

Thank you for this comment. However, we do consider that results need to be addressed in the text in each section (in a brief manner). As this is a qualitative study this is highly relevant as results are not just data but information. This style was adopted in our previous article “Circular Economy Development in the Construction Sector in Japan” and it was considered to be informative.

  1. The data source and method for the entire article are based on document qualitative analysis and questionnaire survey. Can this sample data accurately reflect the overall development of the Finnish timber and construction industry? However, this qualitative data may not necessarily reflect the development trend of the industry in a precise manner.

Reply:

Thank you for this comment. The article reflects the current state of and future outlook on CE development in the wood construction sector in Finland as perceived by various sectoral companies. These companies represent both architectural and construction sector companies. We fully understand the focus and limitations of the study. We have added a sentence about the limitations in conclusions about this. We are also open to a revision of the title if journal editors support that idea and it can be improved.

  1. Overall, the overall reading experience of the entire article appears to be industry report rather than independent research article.

Reply:

Thank you for this comment. However, we do not fully understand this comment as the article is based on qualitative research method and discussion with 133 references. In addition, we have applied this approach in multiple previous scientific articles.

Reviewer 2 Report

Dear Authors,

the paper has been improved following the indications of the reviewers.

The abstract as requested has been improved and the objectives of the work have been highlighted.

The introduction has been expanded with more details on other works already published and the contribution of the authors highlighted.

The conclusions are correct for an article of this type.

In general, the paper is well written and well structured. The research context is clear and the research deserves to be published.

The quality of English is acceptable

Author Response

Dear Authors,

the paper has been improved following the indications of the reviewers.

The abstract as requested has been improved and the objectives of the work have been highlighted.

The introduction has been expanded with more details on other works already published and the contribution of the authors highlighted.

The conclusions are correct for an article of this type.

In general, the paper is well written and well structured. The research context is clear and the research deserves to be published.

Reply:

Thank you for your previous valuable comments to improve the manuscript and it has been improved just the way mentioned above in addition to comments by other reviewers.

Round 3

Reviewer 1 Report

Thank you for the author's prompt response.

I have received answers to some of my questions regarding this article.

Minor editing of English language required